# Disparate Conditional Prediction in Multiclass Classifiers

**Sivan Sabato** [1 2]  **Eran Treister** [2]  **Elad Yom-Tov** [3]

## Abstract

We propose methods for auditing multiclass classifiers for fairness under multiclass equalized odds, by estimating the deviation from equalized odds when the classifier is not completely fair. We generalize to multiclass classifiers the measure of Disparate Conditional Prediction (DCP), originally suggested by Sabato & Yom-Tov (2020) for binary classifiers. DCP is defined as the fraction of the population for which the classifier predicts with conditional prediction probabilities that differ from the closest common baseline. We provide new local-optimization methods for estimating the multiclass DCP under two different regimes, one in which the conditional confusion matrices for each protected sub-population are known, and one in which these cannot be estimated, for instance, because the classifier is inaccessible or because good-quality individual-level data is not available. These methods can be used to detect classifiers that likely treat a significant fraction of the population unfairly. Experiments demonstrate the accuracy of the methods. Code is provided at https://github.com/sivansabato/DCPmulticlass.

## 1. Introduction

Fairness of classifiers is a crucial property in many real-life scenarios (see, e.g., Caton & Haas, 2024). In particular, *auditing* classifiers for fairness is essential in a wide range of applications and has been studied in many works (e.g., Saleiro et al., 2018; Angwin et al., 2022; Taskesen et al., 2021; Cherian & Candès, 2024).

While it is desirable that a classifier accurately satisfies the required fairness criterion, this is in many cases not achiev-

able, due to inherent limitations of the distribution (Pleiss et al., 2017; Kleinberg et al., 2017; Menon & Williamson, 2018; Wang et al., 2024), as well as practical constraints. It is thus necessary to be able to quantify the *deviation* from fairness of a given classifier. In addition, often the classifier is proprietary (Dastin, 2022; Grandinetti, 2023) and thus is not directly accessible for auditing. In other cases, individual-level data that is required for auditing is missing or insufficient. For instance, a health insurance company may use a proprietary classifier to decide on coverage or on premium rates (Kafuria, 2022). Sabato & Yom-Tov (2020) showed that for binary classifiers, it is possible to provide fairness auditing without individual-level data, using only population-level information on the frequencies of positive predictions and of true positive labels in each sub-population, where a sub-population is the group of individuals who have the same value of the protected attribute(s). They considered fairness in the sense of *equalized odds* (Hardt et al., 2016), which defines a binary classifier as fair if its false positive rate and its false negative rate are each the same across all sub-populations. They proposed to quantify the deviation from equalized odds using a measure that we will henceforth call Disparate Conditional Prediction (DCP). DCP is the fraction of the population for which the classifier predicts with conditional prediction probabilities that differ from the closest common baseline.

The fairness notion of equalized odds was originally studied for binary classifiers. However, interest in fairness for *multiclass* classifiers has gained traction in recent years (see, e.g. Alghamdi et al., 2022; Rouzot et al., 2022; Wang et al., 2024). In multiclass scenarios, the multiclass equalized odds criterion measures any differences in conditional prediction probabilities between sub-populations. This includes not only the difference in the rate of correct predictions as in the binary case, but also the types of prediction mistakes. For instance, if a patient's heart attack is misdiagnosed as an anxiety attack (which may mean the patient is denied care), this is significantly different than being misdiagnosed as a stroke (which may lead to delayed care). If some sub-populations incur more of a certain type of misdiagnosis error, this could indicate bias in diagnosis, as well as lead to undesired differences in treatment.

In this work, we study the auditing of multiclass classifiers for deviation from multiclass equalized odds, using

[1]Department of Computing and Software, McMaster University; Canada CIFAR AI Chair, Vector Institute [2]Department of Computer Science, Ben-Gurion University of the Negev [3]Department of Computer Science, Bar-Ilan University. Correspondence to: Sivan Sabato <sabatos@mcmaster.ca>.

*Proceedings of the 42nd International Conference on Machine Learning*, Vancouver, Canada. PMLR 267, 2025. Copyright 2025 by the author(s).

a natural generalization of DCP. DCP is different from other commonly used measures of deviation from equalized odds, such as the difference or ratio of equalized odds (e.g., Alghamdi et al., 2022; Wang et al., 2024), in that it has a consistent interpretable meaning as a fraction of the population, regardless of the number of protected attribute values, the number of classes, or the degree of class imbalance. Thus, DCP is useful for interpretably auditing and comparing classifiers. This is contrasted with the standard use of differences or ratios, which produces undesirable artifacts, such as discounting differences in rare labels, lack of normalization or boundedness, and lack of differentiation between classifiers with different degrees of bias. The quantifiable interpretation of the DCP measure ensures that it does not suffer from similar issues.

It was shown in Sabato & Yom-Tov (2020) that the DCP of a given binary classifier can be calculated efficiently using the confusion matrices of this classifier in each sub-population. When the confusion matrices are not available as discussed above, there exists an efficient procedure for deriving a lower bound on the value of DCP given population-level frequencies. This lower bound can be used to identify classifiers that treat a high fraction of the population unfairly, without direct access to the classifier. In the case of multiclass classifiers, efficient procedures for calculating DCP are unknown, as this may be computationally intractable. Thus, we propose procedures to upper-bound and lower-bound the DCP of a multiclass classifier, given the conditional confusion matrices of the classifier for each sub-population, as well as given only population-level frequencies.

The upper bounds are obtained using local minimization procedures. The minimization problems are constrained, non-smooth and non-convex, and their objective functions have regions with large gradients, which is challenging numerically. As it is known that non-smooth functions are challenging for gradient-based algorithms, we first handle the non-smoothness, caused by a maximum term over several concave functions, by splitting the functions into their smooth parts, and adding more constraints to the existing ones to account for that. Then, we replace the concave functions in the constraints with linear approximations, and the minimization is obtained via sequential solutions of standard linear programming (LP) problems, which have available and rather efficient solution routines (Hall et al., 2023). The LP solvers efficiently handle the constrained minimization at each step, even when the constraint matrices include large numbers because of the large gradients. As the problem is highly non-convex, the sequential minimization reaches local minima.

We report experiments on several data sets, showing that the gap between the upper and lower bounds, for both scenarios, are usually quite small, indicating that the optimization

procedures provide useful estimates. These estimates can be used to identify classifiers that behave differently on different protected sub-populations.

**Paper structure.** Section 2 discusses related work. Preliminaries are provided in Section 3. In Section 4, we present the DCP measure and extend it to multiclass classifiers. Section 5 provides methods for calculating upper and lower bounds for DCP for multiclass classifiers when the conditional confusion matrices for all sub-populations are known. In Section 6, we consider the case where these matrices are unknown, using only population-level frequencies. Experiments are reported in Section 7. We conclude in Section 8. Some technical details are deferred to appendices.

## 2. Related Work

Fairness for multiclass classification has been gaining interest in recent years. Denis et al. (2024) studied multiclass fairness with demographic parity. Alghamdi et al. (2022) use model projection for multiclass equalized odds. Putzel & Lee (2022) propose post-processing techniques for obtaining fairness in multiclass classification for various fairness notions. Rouzot et al. (2022) propose fairness scoring systems for multiclass classification. Wang et al. (2024) study the fundamental limits of fairness in multiclass classifiers, under several fairness notions, including equalized odds.

Sabato & Yom-Tov (2020) studied estimating the (un)fairness of a classifier, a crucial task in many applications (Bellamy et al., 2019). They proposed the new DCP measure for binary classifiers, showed that it is easy to calculate using the confusion matrices for each sub-population, and provided methods for lower-bounding this measure in the absence of individual label data, using population-level frequencies. Several other works have studied fairness auditing in the absence of individual information about protected attributes (e.g., Awasthi et al., 2021; Fabris et al., 2023; Cornacchia et al., 2023). Fairness auditing has also been studied in specific applications, including tax auditing algorithms (Black et al., 2022), visual systems (Goyal et al., 2022), and candidate rankings (Roth et al., 2022).

Many works use some relaxation of equalized odds to allow learning or studying near-fair classifiers. However, there is not a single agreed-upon relaxation, in the binary or the multiclass case. For instance, Donini et al. (2018); Jung et al. (2021); Xue (2023); Wang et al. (2024) use a difference-based formulation, while Calmon et al. (2017) and Alghamdi et al. (2022) use a ratio-based one. In Xue (2023), the sum of the differences is used, while in Wang et al. (2024), the maximum is used. In this work, we study a natural extension to of the interpretable DCP measure of Sabato & Yom-Tov (2020) to multiclass classification.

Our work partly falls within the realm of fairness auditing using only aggregate statistics, without assuming access to the classifier. The challenge of auditing fairness using limited information has received significant attention in recent years, as evident, for example, in (Pinzón et al., 2024; Wang et al., 2021). Our work is unique, in that it is the first, to our knowledge, to address the multiclass setting.

## 3. Preliminaries

We consider a multiclass classification problem with $k$ possible labels, in which each individual in the population has a true label in the label set $\mathcal{Y} \equiv \{1, \ldots, k\}$. Fairness is considered with respect to some protected attribute, such as race, state of residence or age. A value of the attribute, which can be multi-valued, is assigned to each individual. If there is more than one protected attribute, it can be substituted for a single attribute which is the Cartesian product of all protected attributes. A *sub-population* is the subset of the population that includes all the individuals with the same value of the protected attribute.

The object of study is an existing classifier, denote it $\mathcal{C}$, which maps each individual from the population to a predicted label, which may be different from its true label. We do not make any assumptions about the way the classifier is generated or the classification model. For a given $\mathcal{C}$, denote by $\mathcal{D}$ the uniform distribution over the population of the triplets of true label, predicted label, and protected attribute value of individuals. A random triplet drawn according to $\mathcal{D}$ is denoted by $(Y, \hat{Y}, A)$, where $Y \in \mathcal{Y}$ is the true label of the individual, $\hat{Y} \in \mathcal{Y}$ is the label predicted by $\mathcal{C}$ for this individual, and $A \in \mathcal{A}$ is the individual's protected attribute value, where $\mathcal{A}$ is the set of possible values. Denote the probability of an event $E$ according to $\mathcal{D}$ by $\mathbb{P}[E]$.

We define the following notation for properties of $\mathcal{D}$: The frequency of each sub-population in the distribution is $w_a := \mathbb{P}[A = a]$; The vector of frequencies is $\mathbf{w} = (w_a)_{a \in \mathcal{A}}$. The proportion of true label $y \in \mathcal{Y}$ in sub-population $a \in \mathcal{A}$ is $\pi_a^y := \mathbb{P}[Y = y \mid A = a]$; The vector of these proportions is $\boldsymbol{\pi}_a := (\pi_a^y)_{y \in \mathcal{Y}}$. The proportion of predicted label $y \in \mathcal{Y}$ by classifier $\mathcal{C}$ in sub-population $a \in \mathcal{A}$ is $\hat{p}_a^y := \mathbb{P}[\hat{Y} = y \mid A = a]$; The vector of these proportions is $\hat{\mathbf{p}}_a := (\hat{p}_a^y)_{y \in \mathcal{Y}}$. The entries in the confusion matrix of the classifier $\mathcal{C}$ on a sub-population $a \in \mathcal{A}$ are denoted by $\forall y, \hat{y} \in \mathcal{Y}, \alpha_a^{y\hat{y}} := \mathbb{P}[\hat{Y} = \hat{y} \mid Y = y, A = a]$. Throughout this work, the value of a conditional probability expression in which the probability of the condition is zero is treated as zero. Denote the confusion matrix of $\mathcal{C}$ on a sub-population $a$ by $\mathbf{M}_a := (\alpha_a^{y\hat{y}})_{y,\hat{y} \in \mathcal{Y}}$. Denote the indexed set of all such confusion matrices by $\mathbf{M}_{\mathcal{A}} := \{\mathbf{M}_a\}_{a \in \mathcal{A}}$. Denote row $y$ in $\mathbf{M}_a$ by $\boldsymbol{\alpha}_a^y := (\alpha_a^{y\hat{y}})_{\hat{y} \in \mathcal{Y}}$.

Denote the $k$-simplex by $\Delta_k := \{\mathbf{v} \in \mathbb{R}_+^k \mid \|\mathbf{v}\|_1 = 1\}$.

The set of matrices whose rows are in the simplex over $\mathcal{Y}$ is $\Delta_{k \times k} := \{\mathbf{X} := (x^{y\hat{y}})_{y,\hat{y} \in \mathcal{Y}} \mid \forall y \in \mathcal{Y}, (x^{y\hat{y}})_{\hat{y} \in \mathcal{Y}} \in \Delta_k\}$. By definition, the classifier $\mathcal{C}$ satisfies the following:

$$\forall a \in \mathcal{A} \text{ it holds that } \mathbf{M}_a \in \Delta_{k \times k} \text{ and } \mathbf{M}_a^T \boldsymbol{\pi}_a = \hat{\mathbf{p}}_a. \quad (1)$$

The fairness notion of *equalized odds* (Hardt et al., 2016), originally defined for binary classification, was later generalized to multiclass classifiers. We use the term-by-term multiclass equalized odds criterion (Putzel & Lee, 2022; Alghamdi et al., 2022), which requires that for each $y, \hat{y} \in \mathcal{Y}$, $\alpha_a^{y\hat{y}} \equiv \mathbb{P}[\hat{Y} = \hat{y} \mid Y = y, A = a]$ is the same across all $a \in \mathcal{A}$. Equivalently, all the matrices in $\mathbf{M}_{\mathcal{A}}$ are the same. Note that the same definitions can be used also for relaxed multiclass equalized odds criteria, such as those that distinguish sensitive and insensitive labels (see, e.g. Rouzot et al., 2022). This can be achieved by mapping $\hat{Y}$ conditioned $Y = y$ to a smaller set of predicted labels, where the mapping can depend on $y$, and then recalculating the distribution properties provided above. All the methods and results below are valid for the resulting transformation.

## 4. The DCP Measure for Multiclass Classifiers

Recall that the distribution $\mathcal{D}$ is determined by the given classifier $\mathcal{C}$. Denote the conditional distribution of $\mathcal{D}$ given the attribute value $A = a$, and the true label $Y = y$ by $\mathcal{D}_a^y$. The binary DCP measure is based on modeling each $\mathcal{D}_a^y$ as a mixture of two conditional distributions: a global baseline distribution, which is the same for all sub-populations $a$ and represents the baseline behavior, and a local nuisance distribution, which can be different for each $a$ and represents the deviation from this baseline. For $a \in \mathcal{A}, y \in \mathcal{Y}$, let $\eta_a^y \in [0, 1]$ be the probability conditioned on $A = a, Y = y$ that $\hat{Y}$ is drawn according to the nuisance distribution. $\text{DCP}(\mathcal{C})$ is defined as the fraction of the population for which the local nuisance conditional distributions are used by the classifier instead of the global baseline distribution. Since the baseline conditional distribution is unobserved, it is taken to be the one that results in the minimal possible DCP. Formally, for a given classifier $\mathcal{C}$ with the distribution $\mathcal{D}$, and letting $\boldsymbol{\eta}_a := (\eta_a^y)_{y \in \mathcal{Y}}$,

$$\text{DCP}(\mathcal{C}) = \min_{\{\boldsymbol{\eta}_a\}} \sum_{a \in \mathcal{A}} w_a \boldsymbol{\pi}_a^T \boldsymbol{\eta}_a, \quad (2)$$

where the minimum is taken over $\{\boldsymbol{\eta}_a\}_{a \in \mathcal{A}}$ that are consistent with $\mathcal{D}$; that is, such that there exist a baseline distribution $\mathcal{D}_b^y$ and nuisance distributions $\{\mathcal{N}_a^y\}_{y \in \mathcal{Y}, a \in \mathcal{A}}$ such that for each $y \in \mathcal{Y}, a \in \mathcal{A}, \mathcal{D}_a^y = (1 - \eta_a^y)\mathcal{D}_b^y + \eta_a^y \mathcal{N}_a^y$. If the classifier satisfies equalized odds, then the distributions $\mathcal{D}_a^y$ are the same for all $a \in \mathcal{A}$, in which case the decomposition holds by setting $\eta_a^y = 0$ for all $y \in \mathcal{Y}, a \in \mathcal{A}$ and setting the baseline distribution for $y$ to $\mathcal{D}_a^y$ for an arbitrary $a$. This

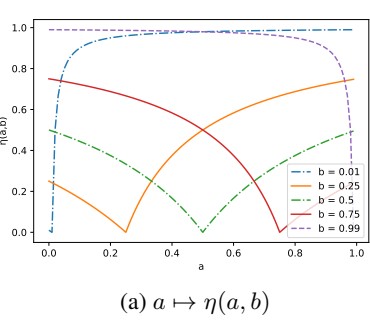

(a) $a \mapsto \eta(a, b)$

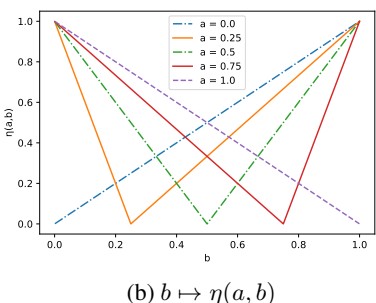

(b) $b \mapsto \eta(a, b)$

Figure 1: The function $\eta(a, b)$.

gives $\text{DCP}(\mathcal{C}) = 0$, as expected. Sabato & Yom-Tov (2020) show that for binary classifiers, where $\mathcal{Y} = \{0, 1\}$,

$$\text{DCP}(\mathcal{C}) = \text{DCP}(\mathbf{M}_{\mathcal{A}}, \mathbf{w}, \boldsymbol{\pi}) := \quad (3)$$
$$\sum_{y \in \mathcal{Y}} \min_{\alpha_{\mathsf{b}}^{y(1-y)} \in [0,1]} \sum_{a \in \mathcal{A}} w_a \pi_a^y \eta(\alpha_{\mathsf{b}}^{y(1-y)}, \alpha_a^{y(1-y)}),$$

where $\mathbf{M}_{\mathcal{A}}$ is determined by $\mathcal{C}$, $\mathbf{w}, \boldsymbol{\pi}$ are the population properties, and $\eta$ without subscripts denotes the function:

$$\eta(a, b) = \begin{cases} 1 - b/a & b < a, \\ 1 - (1-b)/(1-a) & b > a, \\ 0 & b = a. \end{cases} \quad (4)$$

$\eta$ is illustrated in Figure 1. In the minimization in Eq. (3), $\alpha_{\mathsf{b}}^{y(1-y)}$ represent conditional prediction rates for the baseline distribution. More generally, we denote the conditional probability of predicting $\hat{y}$ given true label $y$ under the baseline distribution by $\alpha_{\mathsf{b}}^{y\hat{y}}$. Sabato & Yom-Tov (2020) observe that since $a \mapsto \eta(a, b)$ is piecewise concave on the intervals $[0, b]$ and $[b, 1]$ (see Figure 1), it suffices to minimize each of the terms in Eq. (3) over $\alpha_{\mathsf{b}}^{y(1-y)} \in \{\alpha_a^y\}_{a \in \mathcal{A}} \cup \{0, 1\}$.

We first provide a formulation of $\text{DCP}(\mathbf{M}_{\mathcal{A}}, \mathbf{w}, \boldsymbol{\pi})$ for multiclass classifiers.

**Theorem 4.1.** *For multiclass classification, Eq. (2) implies*

$$\text{DCP}(\mathcal{C}) = \text{DCP}(\mathbf{M}_{\mathcal{A}}, \mathbf{w}, \boldsymbol{\pi}) = \quad (5)$$
$$\sum_{y \in \mathcal{Y}} \min_{\boldsymbol{\alpha}_{\mathsf{b}}^{y} \in \Delta_k} \sum_{a \in \mathcal{A}} w_a \pi_a^y \max_{\hat{y} \in \mathcal{Y}} \eta(\alpha_{\mathsf{b}}^{y\hat{y}}, \alpha_a^{y\hat{y}}),$$

*where $\boldsymbol{\alpha}_{\mathsf{b}}^{y} := (\alpha_{\mathsf{b}}^{y\hat{y}})_{\hat{y} \in \mathcal{Y}}$.*

*Proof.* In an analog to the derivation in Sabato & Yom-Tov (2020), it is easy to see that for a given baseline distribution with rates $\{\alpha_{\mathsf{b}}^{y\hat{y}}\}_{y, \hat{y} \in \mathcal{Y}}$,

$$\alpha_a^{y\hat{y}} = \alpha_{\mathsf{b}}^{y\hat{y}}(1 - \eta_a^y) + \nu_a^{y\hat{y}} \cdot \eta_a^y,$$

where $\nu_a^{y\hat{y}}$ is the conditional probability of predicting $\hat{y}$ given true label $y$, under the nuisance distribution $\mathcal{N}_a^y$. From Eq. (2), it follows that the optimal solution for $\nu_a^{y\hat{y}}$ minimizes $\eta_a^y$ subject to

$$\forall \hat{y} \in \mathcal{Y} \text{ s.t. } \nu_a^{y\hat{y}} \neq \alpha_{\mathsf{b}}^{y\hat{y}}, \eta_a^y = \frac{\alpha_a^{y\hat{y}} - \alpha_{\mathsf{b}}^{y\hat{y}}}{\nu_a^{y\hat{y}} - \alpha_{\mathsf{b}}^{y\hat{y}}},$$

$$\forall y \in \mathcal{Y}, a \in \mathcal{A}, \eta_a^y \geq 0,$$

$$\forall a \in \mathcal{A}, (\nu_a^{y1}, \ldots, \nu_a^{yk}) \in \Delta_k.$$

If $\alpha_a^{y\hat{y}} = \alpha_{\mathsf{b}}^{y\hat{y}}$ for all $\hat{y} \in \mathcal{Y}$, then clearly $\eta_a^y = 0$. Otherwise, the first constraint requires $\nu_a^{y\hat{y}} = \alpha_{\mathsf{b}}^{y\hat{y}} + (\alpha_a^{y\hat{y}} - \alpha_{\mathsf{b}}^{y\hat{y}})/\eta_a^y$. Summing over $\hat{y}$, this implies $\sum_{\hat{y} \in \mathcal{Y}} \nu_a^{y\hat{y}} = \sum_{\hat{y} \in \mathcal{Y}} \alpha_{\mathsf{b}}^{y\hat{y}} + (\sum_{\hat{y} \in \mathcal{Y}} \alpha_a^{y\hat{y}} - \sum_{\hat{y} \in \mathcal{Y}} \alpha_{\mathsf{b}}^{y\hat{y}})/\eta_a^y = 1$. Thus, the last constraint can be replaced by $\nu_a^{y\hat{y}} \in [0, 1]$ for all $\hat{y} \in \mathcal{Y}$. Since the derivative of the RHS of the first constraint is never zero, the minimizer of $\eta_a^y$ is obtained when one of the other constraints holds. That is, with an assignment of $\nu_a^{\hat{y}y} = 1$ for some $a, \hat{y}, y$ such that $\alpha_a^{y\hat{y}} > \alpha_{\mathsf{b}}^{y\hat{y}}$, or $\nu_a^{\hat{y}y} = 0$ for some $a, \hat{y}, y$ such that $\alpha_a^{y\hat{y}} < \alpha_{\mathsf{b}}^{y\hat{y}}$. It follows that there is some $\hat{y} \in \mathcal{Y}$ such that

$$\alpha_a^{y\hat{y}} > \alpha_{\mathsf{b}}^{y\hat{y}} \text{ and } \eta_a^y = \frac{\alpha_a^{y\hat{y}} - \alpha_{\mathsf{b}}^{y\hat{y}}}{1 - \alpha_{\mathsf{b}}^{y\hat{y}}} = 1 - \frac{1 - \alpha_a^{y\hat{y}}}{1 - \alpha_{\mathsf{b}}^{y\hat{y}}}$$

$$\text{or}$$

$$\alpha_a^{y\hat{y}} < \alpha_{\mathsf{b}}^{y\hat{y}} \text{ and } \eta_a^y = \frac{\alpha_a^{y\hat{y}} - \alpha_{\mathsf{b}}^{y\hat{y}}}{-\alpha_{\mathsf{b}}^{y\hat{y}}} = 1 - \frac{\alpha_a^{y\hat{y}}}{\alpha_{\mathsf{b}}^{y\hat{y}}}.$$

Moreover, for all $\hat{y} \in \mathcal{Y}$, if $\alpha_a^{y\hat{y}} > \alpha_{\mathsf{b}}^{y\hat{y}}$ then $\eta_a \geq 1 - \frac{1 - \alpha_a^{y\hat{y}}}{1 - \alpha_{\mathsf{b}}^{y\hat{y}}}$ and if $\alpha_a^{y\hat{y}} < \alpha_{\mathsf{b}}^{y\hat{y}}$ then $\eta_a \geq 1 - \frac{\alpha_a^{y\hat{y}}}{\alpha_{\mathsf{b}}^{y\hat{y}}}$. It follows that

$$\eta_a^y = \max_{\hat{y} \in \mathcal{Y}} \max(1 - \frac{1 - \alpha_a^{y\hat{y}}}{1 - \alpha_{\mathsf{b}}^{y\hat{y}}}, 1 - \frac{\alpha_a^{y\hat{y}}}{\alpha_{\mathsf{b}}^{y\hat{y}}}) = \max_{\hat{y} \in \mathcal{Y}} \eta(\alpha_{\mathsf{b}}^{y\hat{y}}, \alpha_a^{y\hat{y}}).$$

Eq. (5) thus follows. $\square$

It is easy to verify that Eq. (5) is equivalent to Eq. (3) for binary classifiers, since for $\mathcal{Y} = \{0, 1\}$, $\alpha_a^{yy} = 1 - \alpha_a^{y(1-y)}$ and the same for $\alpha_{\mathsf{b}}$, and $\eta(a, b) = \eta(1 - a, 1 - b)$.

For simplicity of notation, we henceforth treat $\mathbf{w}$ and $\boldsymbol{\pi}$ as fixed and write $\text{DCP}(\mathbf{M}_{\mathcal{A}})$. In the multiclass case,

$\text{DCP}(\mathbf{M}_{\mathcal{A}})$ is the solution of a non-convex constrained minimization problem. In the next section, we provide methods for calculating a lower bound and an upper bound for this quantity.

Note that in practice, in some cases, the properties of the distribution $\mathcal{D}$ for $\mathcal{C}$ would be estimated from a limited data set and so may deviate from the true properties for $\mathcal{D}$. However, since DCP is interpretable as a fraction of the population, any inaccuracy in the estimation of $\mathcal{D}$ would map to at most the same amount of inaccuracy in DCP.

## 5. Bounding the DCP of a Multiclass Classifier

We propose methods for calculating a lower bound and an upper bound for $\text{DCP}(\mathbf{M}_{\mathcal{A}})$. The lower bound is a simple analytical formula. For $y \in \mathcal{Y}$, define $\boldsymbol{\alpha}_{\mathcal{A}}^{y} := \{\alpha_a^y\}_{a \in \mathcal{A}}$ and

$$\text{DCP}_y(\boldsymbol{\alpha}_{\mathcal{A}}^y) := \min_{\boldsymbol{\alpha}_b^y \in \Delta_k} \sum_{a \in \mathcal{A}} w_a \pi_a^y \max_{z \in \mathcal{Y}} \eta(\alpha_b^{y\hat{y}}, \alpha_a^{y\hat{y}}). \quad (6)$$

Then $\text{DCP}(\mathbf{M}_{\mathcal{A}}) = \sum_{y \in \mathcal{Y}} \text{DCP}_y(\boldsymbol{\alpha}_{\mathcal{A}}^y)$. A lower bound on $\text{DCP}_y$ can be derived as follows:

$$\begin{aligned}
\text{DCP}_y(\boldsymbol{\alpha}_{\mathcal{A}}^y) &= \min_{\boldsymbol{\alpha}_b^y \in \Delta_k} \sum_{a \in \mathcal{A}} w_a \pi_a^y \max_{\hat{y} \in \mathcal{Y}} \eta(\alpha_b^{y\hat{y}}, \alpha_a^{y\hat{y}}) \quad (7) \\
&\geq \min_{\boldsymbol{\alpha}_b^y \in [0,1]^k} \sum_{a \in \mathcal{A}} w_a \pi_a^y \max_{\hat{y} \in \mathcal{Y}} \eta(\alpha_b^{y\hat{y}}, \alpha_a^{y\hat{y}}) \\
&\geq \min_{\boldsymbol{\alpha}_b^y \in [0,1]^k} \max_{\hat{y} \in \mathcal{Y}} \sum_{a \in \mathcal{A}} w_a \pi_a^y \eta(\alpha_b^{y\hat{y}}, \alpha_a^{y\hat{y}}) \\
&= \max_{\hat{y} \in \mathcal{Y}} \min_{x \in [0,1]} \sum_{a \in \mathcal{A}} w_a \pi_a^y \eta(x, \alpha_a^{y\hat{y}}).
\end{aligned}$$

Due to the piecewise concavity of $a \mapsto \eta(a, b)$ on $[0, b]$ and $[b, 1]$, it suffices to minimize each of the terms in Eq. (3) over $x \in \{0, 1\} \cup \{\alpha_a^{y\hat{y}}\}_{a \in \mathcal{A}}$. Thus,

$$\text{DCP}(\mathbf{M}_{\mathcal{A}}) \geq \sum_{y \in \mathcal{Y}} \max_{\hat{y} \in \mathcal{Y}} \min_{x \in \{0,1\} \cup \{\alpha_a^{y\hat{y}}\}_{a \in \mathcal{A}}} \sum_{a \in \mathcal{A}} w_a \pi_a^y \eta(x, \alpha_a^{y\hat{y}}).$$

This lower bound can be calculated in time $O(|\mathcal{A}|^2 k^2)$.

To upper bound $\text{DCP}(\mathbf{M}_{\mathcal{A}})$, we propose a local iterative optimization approach on the objective function in Eq. (6), which is constrained, non-smooth, and non-convex. One can randomly initialize using a feasible solution. However, we propose a more tailored approach at the end of this section.

Our optimization approach utilizes sequential LP solutions, and is as follows. Given $y \in \mathcal{Y}$, denote the matrix $\mathbf{H} \in \mathbb{R}^{|\mathcal{A}| \times k}$ to include the entries $h_{a,\hat{y}} := \alpha_a^{y\hat{y}}$. Note that the rows of $\mathbf{H}$ are in the simplex $\Delta_k$. The vector $\widetilde{\mathbf{w}} \in [0, 1]^{|\mathcal{A}|}$ includes the entries $\widetilde{w}_a := w_a \pi_a^y$. The optimization variables are denoted by the vector $\boldsymbol{\alpha} := \boldsymbol{\alpha}_b^y \in \mathbb{R}^k$. Denote the all-one vector of dimension $d$ by $\mathbf{1}_d$ and the

**Algorithm 1** Local minimization of $\text{DCP}_y$ via sequential linear programming

---
**Input:** $\boldsymbol{\alpha}^{(0)}, \mathbf{H}, \widetilde{\mathbf{w}}, \text{maxIter}, \varepsilon$
**Output:** The local minimizer $\boldsymbol{\alpha}^*$
**for** $t = 0$ **to** maxIter **do**
 Define the LP approximation of Eq. (9) by computing $\mathbf{M}_1, \mathbf{M}_2$, and $\mathbf{v}$.
 Define $\hat{\boldsymbol{\alpha}}$ as the minimizer of the LP in Eq. (11).
 Set $\boldsymbol{\alpha}^{(t+1)} = \boldsymbol{\alpha}^{(t)} + \mu(\hat{\boldsymbol{\alpha}} - \boldsymbol{\alpha}^{(t)})$, where $\mu$ is obtained using line search to ensure a reduction in Eq. (8).
 If $\|\mathbf{e}\| < \varepsilon$, stop.
**end for**
**return** the last iterate $\boldsymbol{\alpha}^{(t)}$.

---

identity matrix of dimensions $d \times d$ by $\mathbf{I}_d$. Our objective in Eq. (5) and its constraints, is thus given by:

$$\underset{\boldsymbol{\alpha} \in \mathbb{R}^k}{\text{Minimize}} \quad \sum_a \widetilde{w}_a \max_{\hat{y}} \{\eta(\alpha_{\hat{y}}, h_{a,\hat{y}})\} \quad (8)$$

$$\text{s.t.} \quad 0 \leq \boldsymbol{\alpha} \leq 1,$$
$$\langle \boldsymbol{\alpha}, \mathbf{1}_k \rangle = 1.$$

To solve the problem, we observe that its structure resembles a linear program (LP). That is, the objective is given by an inner product, and the constraints are linear. The only difference between an LP and Eq. (8) is the maximum term over the $\eta$ values inside the inner product. Hence, to solve Eq. (8), we apply the sequential linear programming approach (Nocedal & Wright, 2006), which iteratively approximates Eq. (8) as a linear program and solves it.

First, we replace the maximum terms in the objective with another variable vector $\mathbf{c}$ and additional constraints, following Charalambous & Conn (1978):

$$\underset{\boldsymbol{\alpha} \in \mathbb{R}^k, \mathbf{c} \in \mathbb{R}^{|\mathcal{A}|}}{\text{Minimize}} \quad \langle \widetilde{\mathbf{w}}, \mathbf{c} \rangle \quad (9)$$

$$\text{s.t.} \quad 0 \leq \boldsymbol{\alpha}, \mathbf{c} \leq 1,$$
$$\langle \boldsymbol{\alpha}, \mathbf{1}_k \rangle = 1,$$
$$\eta(\alpha_{\hat{y}}, h_{a,\hat{y}}) \leq c_a, \forall \hat{y} \in [k], \forall a \in \mathcal{A}.$$

This problem is equivalent to Eq. (8), but has no maximum operation. However, now we have non-linear constraints, thus violating the definition of an LP. To correct this, we use a local approximation of $\eta$ (around an iterate $\boldsymbol{\alpha}^{(t)}$) using the Taylor series. Note that the function $a \mapsto \eta(a, b)$ is piecewise concave, non-negative, and smooth at $(0, 1)$ except at $b$, where it is minimized and equals zero. Therefore, we do not expect the maximum over $\hat{y}$ in the objective Eq. (8) to fall on the discontinuity of $\eta$ at $b$, except for extreme cases. Hence, using a linear approximation we expect to get a good local approximation of $\eta()$.

The sequential linear programming approach calculates the new iterate $(\boldsymbol{\alpha}^{(t+1)}, \mathbf{c}^{(t+1)})$ by solving an LP problem of an

approximation of Eq. (9) around $(\boldsymbol{\alpha}^{(t)}, \mathbf{c}^{(t)})$. That is, given an iterate $(\boldsymbol{\alpha}^{(t)}, \mathbf{c}^{(t)})$ we first approximate $\eta$ by a linear Taylor series in its first argument:

$$\eta(\alpha+\epsilon, b) - \eta(\alpha, b) \approx \frac{\partial\eta(\alpha, b)}{\partial\alpha}\epsilon = \begin{cases} \frac{b}{\alpha^2}\epsilon & \alpha > b, \\ \frac{-(1-b)}{(1-\alpha)^2}\epsilon & \alpha < b, \\ 0 & \alpha = b. \end{cases}$$

We now locally approximate Eq. (9) around an iterate $\boldsymbol{\alpha}^{(t)}$ via the following LP problem:

$$\underset{\boldsymbol{\alpha}\in\mathbb{R}^k, \mathbf{c}\in\mathbb{R}^{|\mathcal{A}|}}{\text{Minimize}} \quad \langle\widetilde{\mathbf{w}}, \mathbf{c}\rangle \tag{10}$$

$$\text{s.t.} \quad 0 \le \boldsymbol{\alpha}, \mathbf{c} \le 1,$$
$$\langle\boldsymbol{\alpha}, \mathbf{1}_k\rangle = 1,$$
$$\mathbf{v}_a + \mathbf{J}_a(\boldsymbol{\alpha} - \boldsymbol{\alpha}^{(t)}) \le \mathbf{1}_k c_a, \forall a \in \mathcal{A},$$

where $\mathbf{v}_a \in \mathbb{R}^k$ is a vector with the coordinates $(\mathbf{v}_a)_{\hat{y}} = \eta(a_{\hat{y}}^{(t)}, h_{a,\hat{y}})$ for all $\hat{y} \in [k]$, and $\mathbf{J}_a \in \mathbb{R}^{k \times k}$ is a diagonal Jacobian matrix s.t $(\mathbf{J}_a)_{\hat{y}\hat{y}} = \frac{\partial\eta(a_{\hat{y}}^{(t)}, h_{a,\hat{y}})}{\partial a}$. Bringing this into a canonical form with a variable $\boldsymbol{\alpha} = \boldsymbol{\alpha}^{(t)} + \mathbf{e}$ yields

$$\underset{\boldsymbol{\alpha}\in\mathbb{R}^k, \mathbf{c}\in\mathbb{R}^{|\mathcal{A}|}}{\text{Minimize}} \quad \langle\widetilde{\mathbf{w}}, \mathbf{c}\rangle \tag{11}$$

$$\text{s.t.} \quad 0 \le \boldsymbol{\alpha}, \mathbf{c} \le 1,$$
$$\langle\boldsymbol{\alpha}, \mathbf{1}_k\rangle = 1,$$
$$\mathbf{M}_1\mathbf{c} + \mathbf{M}_2\boldsymbol{\alpha} - \mathbf{M}_2\boldsymbol{\alpha}^{(t)} + \mathbf{v} \le 0,$$

where $\mathbf{M}_1 := -\mathbf{I}_{|\mathcal{A}|} \otimes \mathbf{1}_k$, $\mathbf{M}_2 := [\mathbf{J}_1; \mathbf{J}_2; ...; \mathbf{J}_{|\mathcal{A}|}]$ is the stacking of all diagonal Jacobian matrices on top of each other, and $\mathbf{v} := [\mathbf{v}_1; ...; \mathbf{v}_{|\mathcal{A}|}]$ is the stacking of vectors. Eq. (11) is an LP in a canonical form, which is solved iteratively. Note that $\mathbf{M}_1$ and $\mathbf{M}_2$ are highly sparse, which can be exploited to easily solve large instances of the problem. The resulting local optimization algorithm is given in Alg. 1.

**A note on usage.** By definition, the function $\eta$ in Eq. (4) is continuous in the open section $(0, 1)^2$, but at the boundaries, it is easy to see that $\eta(0, 0) = 0$, while $\lim_{\delta\to 0}\eta(\delta, 0) = 1$. Furthermore, the derivatives of $\eta$ when its first argument is close to 0 or 1 can be very large, causing numerical difficulties in the LP solvers. Thus, to avoid numerical instability, we make sure that all entries of $\mathbf{H}$ are in $[\varepsilon, 1-\varepsilon]$ (where we used $\varepsilon = 10^{-5}$) by replacing any value outside these boundaries by the relevant boundary value and renormalizing. Note that since the input values in $\mathbf{H}$ are calculated in practice based on finite data sets, values in $\mathbf{H}$ are already noisy representations of the true population values, and so slightly changing them does not negatively affect the correctness of the method.

**Finding an initializing assignment** A natural guess for an initializing assignment $\boldsymbol{\alpha}^{(0)}$ is the weighted average of the confusion matrices in $\mathbf{M}_{\mathcal{A}}$. However, this does not take

the objective function into account. We propose instead a greedy approach, in which the entries in the baseline confusion matrix are optimized label-by-label, using the fact that the binary problem is easy to minimize. In the first iteration, all labels except for one are treated as the same label, and optimal confusion matrix values for the first labels are calculated. In each further iteration, one additional label is separated, and the assignment for this label is optimized given the assignments of the previous labels. This approach is possible because DCP is monotonic under the merging of labels. The full derivation and procedure are provided in Appendix C. Our experiments in Section 7 demonstrate that this method is superior to the averaging approach.

## 6. Best-case DCP without Confusion Matrices

We next consider the case of a classifier for which the confusion matrices for each sub-population are not available, either because the classifier is not accessible for testing or because individual-level ground-truth data is missing or insufficient. In this case, we assume, as in Sabato & Yom-Tov (2020), access only to the frequencies of true and predicted labels in each sub-population, but not the conditional probabilities. Formally, we do not have $\mathbf{M}_{\mathcal{A}}$ for the given classifier $\mathcal{C}$. We only have access to the population-level frequencies $(\mathbf{w}, \{(\boldsymbol{\pi}_a, \hat{\mathbf{p}}_a)\}_{a\in\mathcal{A}})$.

If $\mathcal{C}$ is fair under multiclass equalized odds, then all the matrices in $\mathbf{M}_{\mathcal{A}}$ must be identical. Thus, from Eq. (1), there exists a single confusion matrix $\mathbf{M} \in \Delta_{k\times k}$ such that $\forall a \in \mathcal{A}, \mathbf{M}^T\boldsymbol{\pi}_a = \hat{\mathbf{p}}_a$. If such a matrix does not exist and the input frequencies are accurate, then $\mathcal{C}$ does not satisfy the multiclass equalized odds criterion. Nonetheless, given $(\mathbf{w}, \{(\boldsymbol{\pi}_a, \hat{\mathbf{p}}_a)\}_{a\in\mathcal{A}})$ for $\mathcal{C}$, we would like to find the best-case value of DCP for $\mathcal{C}$, denoted minDCP. Formally,

$$\texttt{minDCP}((\mathbf{w}, \{(\boldsymbol{\pi}_a, \hat{\mathbf{p}}_a)\}_{a\in\mathcal{A}})) := \tag{12}$$
$$\min\{\text{DCP}(\mathbf{M}_{\mathcal{A}}) \mid \forall a \in \mathcal{A}, \mathbf{M}_a \in \Delta_{k\times k}, \mathbf{M}_a^T\boldsymbol{\pi}_a = \hat{\mathbf{p}}_a\}.$$

Sabato & Yom-Tov (2020) show that minDCP can be calculated exactly for binary classifiers. In the multiclass case, we are not aware of an efficient algorithm for calculating the exact value of minDCP. Instead, we provide below a local-minimization algorithm for $\text{DCP}(\mathbf{M}_{\mathcal{A}})$ under the constraints on $\mathbf{M}_{\mathcal{A}}$, which results in an upper bound for minDCP. This, combined with a lower bound on minDCP, provides a limited range of possible values of minDCP for the given classifier $\mathcal{C}$.

To calculate the lower bound, observe that

$$\text{DCP}(\mathbf{M}_{\mathcal{A}}) = \sum_{y\in\mathcal{Y}} \min_{\boldsymbol{\alpha}_b^y\in\Delta_k} \sum_{a\in\mathcal{A}} w_a\pi_a^y \max_{z\in\mathcal{Y}} \eta(\alpha_b^{y\hat{y}}, \alpha_a^{y\hat{y}})$$
$$\ge \sum_{y\in\mathcal{Y}} \min_{\alpha_b^{yy}\in[0,1]} \sum_{a\in\mathcal{A}} w_a\pi_a^y \eta(\alpha_b^{yy}, \alpha_a^{yy}). \tag{13}$$

Each of the terms in the RHS is of the same form as the terms in the definition of DCP for binary classification. The RHS is a constrained minimization problem that can be solved using the same methodology as in Sabato & Yom-Tov (2020), providing a lower bound for `minDCP`.

We now turn to the local optimization algorithm for `minDCP`. We use a sequential linear programming approach, similarly to our solution to Eq. (8) in Section 5. First, we extend the notation of Eq. (8) to explicitly denote the value of $y \in \mathcal{Y}$: The matrix $\mathbf{H}^y \in \mathbb{R}^{|\mathcal{A}| \times k}$ is defined such that $h_{a\hat{y}}^y = \alpha_a^{y\hat{y}}$; The vector $\widetilde{\mathbf{w}}^y \in [0,1]^{|\mathcal{A}|}$ includes the entries $\widetilde{w}_a^y := w_a \pi_a^y$; The optimization variables are denoted by the vectors $\boldsymbol{\alpha}^y := \boldsymbol{\alpha}_b^y \in \mathbb{R}^k$. We replace the maximum terms in the problem in Eq. (5) with the vectors $\mathbf{c}^y \in \mathbb{R}^k$, similarly to the transformation from Eq. (8) to Eq. (9). The variables for the optimization problem are the collection of the vectors and matrices

$$\mathbf{x} = \{\boldsymbol{\alpha}^1, \mathbf{H}^1, \mathbf{c}^1, ..., \boldsymbol{\alpha}^k, \mathbf{H}^k, \mathbf{c}^k\}, \qquad (14)$$

which amount to $k^2 + k^2 \cdot |\mathcal{A}| + k \cdot |\mathcal{A}|$ scalar unknowns in total. We get the following new problem, which is equivalent to the problem defined in Eq. (5), including its constraints:

$$\underset{\{\boldsymbol{\alpha}^y\},\{\mathbf{H}^y\},\{\mathbf{c}^y\}}{\text{Minimize}} \quad \sum_y \langle \widetilde{\mathbf{w}}^y, \mathbf{c}^y \rangle \qquad (15)$$

$$\text{s.t.} \quad 0 \leq \boldsymbol{\alpha}^y \leq 1 \quad \forall y \in [k],$$
$$0 \leq \mathbf{H}^y \leq 1 \quad \forall y \in [k],$$
$$0 \leq \mathbf{c}^y \leq 1 \quad \forall y \in [k],$$
$$\langle \boldsymbol{\alpha}^y, 1 \rangle = 1 \quad \forall y \in [k],$$
$$\mathbf{H}^y \mathbf{1}_k = \mathbf{1}_{|\mathcal{A}|} \quad \forall y \in [k],$$
$$\sum_{y=1}^k \pi_a^y h_{a,\hat{y}}^y = p_a^{\hat{y}} \quad \forall \hat{y} \in [k], a \in [|\mathcal{A}|].$$
$$\eta(\alpha_{\hat{y}}^y, h_{a,\hat{y}}^y) \leq c_a^y \quad \forall y, \hat{y} \in [k], \ a \in [|\mathcal{A}|],$$

Note that the objective in Eq. (15) is linear, and its constraints are linear as well, except the bottom ones involving $\eta()$, similarly to Eq. (9). Also note that all of the linear constraints are separable in $y$, except for the last one, which introduces a coupling between the $\mathbf{H}^y$ of different $y$'s.

To solve the problem we again use the sequential linear programming approach, as the structure of the problem again resembles a linear program. Given the $t$-th iterate $\mathbf{x}^{(t)}$ for all the variables defined in Eq. (14), we first approximate $\eta$ by a linear Taylor series, this time in both of its arguments, since now both are optimized. The function $\eta(\alpha, b)$ is linear in $b$, hence the Taylor series is exact with respect to $b$, which is appealing for our approach. However, minimizing the objective with respect to both $h_{a,\hat{y}}^y$ and $\alpha_a^y$ leads to the minimal non-smooth points of $\eta$. These points are the singularity points (see Fig. 1), where $\eta$ is not approximated well by a linear function. This results in a slow convergence of the sequential LP algorithm. To solve this, we split $\eta$ and define

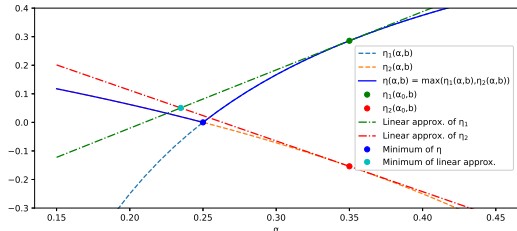

Figure 2: The linear approximation of $\eta(\alpha, b)$ using the split to $\eta_1$ and $\eta_2$ around $\alpha_0 = 0.35$, for a fixed $b = 0.25$.

it as the maximum between the two smooth functions:

$$\eta(\alpha, b) = \max\{\eta_1(\alpha, b), \eta_2(\alpha, b)\}, \qquad (16)$$

where $\eta_1(\alpha, b) = 1 - \frac{b}{\alpha}$ and $\eta_2(\alpha, b) = 1 - \frac{1-b}{1-\alpha}$. This split is an alternative formulation to the definition of $\eta$ in Eq. (4), only now the singularity point is replaced with a maximum, and can be treated in the same way we treat the maximization over all terms involving $\eta$ using sequential LP. Figure 2 shows the independent linearization of the two terms after the split and the piece-wise linear approximation of the non-smooth function in our objective. Overall, the solution of the Eq. (15) is obtained after the split using sequential LP. More details are given in Appendix D.

## 7. Experiments

We report experiments demonstrating the performance of the methods proposed above, for the case of known as well as unknown confusion matrices (Section 7.1 and Section 7.2), respectively. We show that in most cases, the ratio between the lower bound and the upper bound, in both types of experiments, is close to 1, indicating that the methods produce fairly accurate estimates. Code is provided at https://github.com/sivansabato/DCPmulticlass.

### 7.1. Bounding the DCP with Known Confusion Matrices

We report the results of calculating an upper bound and a lower bound on the $\text{DCP}(\mathbf{M}_{\mathcal{A}})$ of a multiclass classifier given $\mathbf{M}_{\mathcal{A}}$. As discussed above, if the confusion matrices in $\mathbf{M}_{\mathcal{A}}$ are known up to some accuracy of $\mathcal{D}$, then $\text{DCP}(\mathbf{M}_{\mathcal{A}})$ will have at least the same accuracy, as a direct result of its definition as a fraction of the population.

We compared our proposed local-optimization approach for calculating an upper bound, described in Section 5, to other approaches for finding a feasible low-value assignment for Eq. (5). We tested the following alternatives: (1) Average: the weighted average of the sub-population confusion matrices; (2) Greedy: the greedy initialization procedure proposed in Section 5; (3) Average+LM and (4) Greedy+LM:

initializing with the corresponding method, then running the local optimization procedure. Our results below show that our approach, Greedy+LM, has the best performance. We also report the ratio between the best upper bound and the lower bound, calculated according to Eq. (7). A ratio close to 1 indicates that in this case, the two bounds are tight.

For these experiments, we selected data sets on individuals, which are mostly categorical, and which include several multi-valued attributes that can be used as a predicted label and as a protected attribute. First, we used the US Census data set (Dua & Graff, 2019) to generate multiclass classifiers to predict each of the individual attributes that have between 3 and 10 values. The protected attribute was the state of work. We tested our methods with three different types of classifiers: a decision tree and a nearest neighbor classifier, using standard Matlab libraries, as well as a classifier based on a standard fully connected neural network. The latter used two layers of size 64 and 32 neurons, with ReLU activation. The network was implemented in PyTorch, and trained with the AdamW optimizer, with a learning rate of 0.001, and a batch size of 256, for 4 epochs.

Table 1, as well as Table 3 and Table 5 in Appendix A, list the DCP lower bound calculated for each of these classifiers given their confusion matrices, the upper bound obtained by each of the tested upper bound methods, and the ratio between the smallest upper bound and the lower bound. In some of the experiments, the lower bound is equal to the upper bound, giving the exact value of the DCP of this classifier. In all of the experiments except for a single case, our proposed approach provides the tightest upper bound. The ratio between the upper bound and the lower bound ranges from 1.00 to 2.85, and is close to 1.00 in most cases, showing that for these classifiers, the DCP can be estimated quite well. In most cases, the DCP for decision tree classifiers obtained better accuracy ratios.

Second, we used data about births in the United States (CDC, 2017), which provides detailed information about each birth that occurred during 2017. The data set includes approximately 3.8 million data points and 50 attributes. Using only attributes that are known before the labor, we generated three classifiers that attempt to predict the type of labor out of the five possible options (e.g., spontaneous, Cesarean): a decision tree (error 30.8%), a $k$-Nearest Neighbor classifier, where $k$ was set to 9 using parameter tuning (error 24.6%), and a 2-layer neural network (error 22.43%). This allows studying a high-error regime for the estimation of DCP. We estimated the DCP of each classifier, with respect to different protected attributes. Table 1 (bottom), and Table 4, Table 6 in Appendix A, report the DCP lower bound and upper bounds relative to each of these protected attributes. Here too, our proposed approach provides the tightest upper bounds. The ratio between the upper bound and the

Table 1: DCP with known confusion matrices, decision trees; US Census (top), Natality (bottom). For US Census, each row corresponds to a different classifier. Its error is indicated in the "Error" column. "Lower bound" is the lower bound calculated for DCP using our method. The upper bounds are the values obtained by each of the compared methods for minimizing the DCP objective. "Best ratio" indicates the ratio between the best (lowest) upper bound and the lower bound. A ratio of 1 indicates that the bounds are both tight. For Natality, all the rows report the same classifier, and each row calculates DCP with respect to a different protected attribute.

| # Labels | Error | Lower Bound | Upper Bounds | | | | Best Ratio |
|---|---|---|---|---|---|---|---|
| | | | Average | Greedy | Average+LM | Greedy+LM | |
| 3 | 11.74% | 5.39% | 27.19% | 9.65% | 14.07% | **7.65%** | 1.42 |
| 3 | 5.71% | 4.35% | 42.17% | 5.92% | 32.39% | **5.28%** | 1.21 |
| 3 | 3.96% | 3.24% | 43.63% | 5.07% | 16.95% | **3.25%** | 1.00 |
| 3 | 5.15% | 4.24% | 39.05% | 5.40% | 14.32% | **4.25%** | 1.00 |
| 3 | 3.36% | 2.65% | 48.04% | 5.20% | 5.49% | **3.81%** | 1.44 |
| 3 | 1.85% | 1.85% | 59.64% | 8.22% | 17.85% | **1.85%** | 1.00 |
| 3 | 1.96% | 1.96% | 51.86% | 7.88% | 13.31% | **1.96%** | 1.00 |
| 3 | 2.32% | 2.28% | 48.50% | 6.57% | 10.56% | **2.28%** | 1.00 |
| 3 | 14.00% | 4.57% | 28.85% | 6.47% | 13.13% | **6.10%** | 1.34 |
| 4 | 3.91% | 1.48% | 27.55% | 8.12% | 1.86% | **1.49%** | 1.00 |
| 5 | 5.61% | 2.06% | 7.14% | 43.01% | 6.61% | **3.91%** | 1.90 |
| 5 | 11.83% | 4.57% | 25.28% | 34.40% | 22.18% | **8.28%** | 1.81 |
| 6 | 0.87% | 0.86% | 21.96% | 24.65% | 9.55% | **0.86%** | 1.00 |
| 8 | 22.61% | 8.29% | 84.54% | 32.03% | 38.20% | **23.61%** | 2.85 |
| 9 | 21.54% | 5.03% | 86.17% | 12.50% | 6.84% | **6.17%** | 1.23 |

| Protected Attribute | Lower Bound | Upper Bounds | | | | Best Ratio |
|---|---|---|---|---|---|---|
| | | Average | Greedy | Average+LM | Greedy+LM | |
| Attendant | 1.91% | 14.59% | 2.34% | 2.10% | **2.08%** | 1.09 |
| Father Race | 0.92% | 19.75% | 1.50% | 1.32% | **1.28%** | 1.40 |
| Mother Race | 0.65% | 13.43% | 1.31% | 1.14% | **1.12%** | 1.71 |
| Payer | 1.74% | 24.15% | 1.96% | 1.97% | **1.89%** | 1.09 |

lower bound is between 1.00 and 1.58, showing that here too, DCP is estimated to a high accuracy.

### 7.2. Bounding `min`DCP without Confusion Matrices

In the second set of experiments, we used only the population-level frequencies $(\mathbf{w}, \{(\boldsymbol{\pi}_a, \hat{\mathbf{p}}_a)\}_{a \in \mathcal{A}})$ to estimate `minDCP`. We ran the local optimization procedure provided in Section 6 to find an upper bound on `minDCP`, and compared this result to the lower bound Eq. (13), to provide a range of possible values for `minDCP`. This range can indicate whether the population-level frequencies point to a possible, or definite, large deviation from multiclass equalized odds, as measured by the DCP.

First, we calculated the local minimizer of `minDCP` for the same two labeled data sets that were tested in Section 7.1, this time without access to the confusion matrices. We then compared the obtained value to the actual ranges of DCP that were calculated using the confusion matrices in Section 7.1. Table 2 provides results for the US Census data set and the Natality data set. In all but a single case the local optimizer of `minDCP` was lower than the true DCP range of the classifier, showing that this value is indeed a relevant best-case value for DCP with unknown confusion matrices.

Next, we report two experiments for which we do not have ground truth labels to compare to, to demonstrate how this

Table 2: Comparing the output of the `minDCP` local optimizer (LO) to the DCP range calculated for US Census classifiers (top) and Natality (bottom). The ranges are derived from the results of Table 1, Table 3 and Table 4.

| US Census | Decision tree | | Nearest Neighbor | |
|---|---|---|---|---|
| # Labels | `minDCP` LO | true DCP | `minDCP` LO | true DCP |
| 3 | 2.47% | 5.39% − 7.65% | 3.16% | 6.56% − 9.77% |
| 3 | 1.11% | 4.35% − 5.28% | 2.86% | 6.13% − 9.87% |
| 3 | 1.20% | 3.24% − 3.25% | 3.01% | 6.82% − 9.59% |
| 3 | 0.84% | 4.24% − 4.25% | 2.79% | 6.16% − 8.81% |
| 3 | 0.98% | 2.65% − 3.81% | 2.83% | 7.15% − 9.05% |
| 3 | 1.13% | 1.85% | 2.82% | 6.93% − 9.08% |
| 3 | 1.11% | 1.96% | 3.02% | 6.56% − 8.59% |
| 3 | 0.68% | 2.28% | 2.82% | 7.14% − 9.47% |
| 3 | 1.99% | 4.57% − 6.10% | 2.40% | 5.24% − 7.01% |
| 4 | 1.07% | 1.48% − 1.49% | 4.29% | 7.53% − 10.81% |
| 5 | 2.08% | 2.06% − 3.91% | 6.03% | 8.86% − 10.74% |
| 5 | 3.13% | 4.57% − 8.28% | 3.29% | 8.86% − 10.74% |
| 5 | – | – | 5.99% | 9.50% − 18.22% |
| 5 | – | – | 5.62% | 8.88% − 21.25% |
| 6 | 0.32% | 0.86% | 6.41% | 10.10% − 20.58% |
| 8 | 4.08% | 8.29% − 23.61% | 5.97% | 7.89% − 20.45% |
| 9 | 2.38% | 5.03% − 6.17% | 4.31% | 7.91% − 14.09% |

| Natality | Decision tree | | $k$-Nearest-Neighbors | |
|---|---|---|---|---|
| Protected Attribute | `minDCP` LO | true DCP | `minDCP` LO | true DCP |
| Attendant | 0.75% | 1.91% − 2.08% | 0.74% | 1.80% − 1.82% |
| Father Race | 0.26% | 0.92% − 1.28% | 0.32% | 1.17% − 1.18% |
| Mother Race | 0.12% | 0.65% − 1.12% | 0.14% | 0.61% − 0.62% |
| Payer | 0.05% | 1.74% − 1.89% | 0.59% | 1.73% − 1.75% |

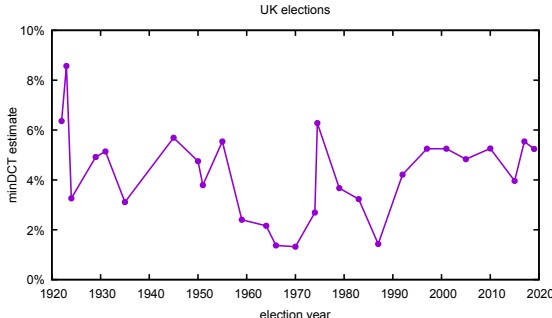

Figure 3: Calculated value `minDCP` for each election year in the UK elections dataset, where the value is reported for the classifier that attempts to predict this year's election result using the results of the previous election.

method can be used for studying various empirical questions. In the first experiment, we used data about the general elections in the UK from 1918 to 2019 (Watson et al., 2020). We studied the changes in voting patterns between elections by studying the `minDCP` value of a hypothetical classifier that would predict the vote of individuals in one general election to be the same as their vote in the previous general elections (ignoring the change in population between elections). We studied DCP when the protected attribute was the geographic region of the voters, as reported in the data set. A high DCP value of such a classifier would indicate a possibly large difference between voting pattern changes across regions. Figure 3 shows the value of `minDCP` by election year, revealing clear differences between different periods of the last century.[1] The full results are reported in Table 9 in the appendix. In Appendix B, we report an additional experiment, on a US education data set.

## 8. Conclusion

In this work, we provided a definition and bounding methods for the DCP measure for multiclass classifiers. This provides a new tool for auditing fairness in multiclass classification. Because DCP is interpretable as a fraction of the population, the estimation methods that we proposed can be used to provide a clear evaluation of classifiers' deviation

---

[1]Note the spike in Figure 3 for Oct. 1974 elections; they were unusual as they were held in the same year as the previous elections, and resulted in a significant political change (Roe-Crines, 2021).

from equalized odds, even when there are many classes, protected attribute values, or highly imbalanced data sets.

## Impact Statement

This paper studies the auditing of classifiers for fairness in the sense of multiclass equalized odds. Identifying classifiers that may violate fairness criteria is an important task that can help advance societal desiderata. Nonetheless, such use must always consider the suitability of the fairness criterion to the specific classification task. In addition, in some cases the methods that we propose may provide loose bounds for the studied criterion. These cases can be identified by observing a large gap between the upper bound and the lower bound. In such cases, care must be taken when making operative decisions based on the results of the methods.

## Acknowledgements

We acknowledge the support of the Natural Sciences and Engineering Research Council of Canada (NSERC), [funding reference number RGPIN-2024-05907]. Resources used in preparing this research were provided, in part, by the Province of Ontario, the Government of Canada through CIFAR, and companies sponsoring the Vector Institute; see https://vectorinstitute.ai/partnerships/current-partners/. This work was supported by the Israeli Council for Higher Education (CHE) via the Data Science Research Center at Ben-Gurion University.

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

## A. Deferred Experiment Results

Table 3—Table 9 below provide the remaining result of the experiments which are described in Section 7.

Table 3: Bounding the DCP for multiclass classifiers with known confusion matrices; US Census data set with nearest neighbor classifiers. Table columns as in Table 1.

| #      | Error   | Lower   | Upper Bounds | | | | Best  |
|--------|---------|---------|---------|---------|------------|-----------|-------|
| Labels |         | Bound   | Average | Greedy  | Average+LM | Greedy+LM | Ratio |
| 3      | 27.71%  | 6.56%   | 19.71%  | 10.64%  | 10.80%     | **9.77%**  | 1.49  |
| 3      | 28.71%  | 6.13%   | 19.00%  | 11.66%  | 10.67%     | **9.87%**  | 1.61  |
| 3      | 26.22%  | 6.82%   | 18.44%  | 10.46%  | 11.82%     | **9.59%**  | 1.41  |
| 3      | 24.61%  | 6.16%   | 19.54%  | 10.05%  | 11.23%     | **8.81%**  | 1.43  |
| 3      | 24.24%  | 7.15%   | 20.04%  | 10.44%  | 12.04%     | **9.05%**  | 1.27  |
| 3      | 23.52%  | 6.93%   | 20.42%  | 10.00%  | 11.97%     | **9.08%**  | 1.31  |
| 3      | 23.43%  | 6.56%   | 19.15%  | 8.91%   | 11.10%     | **8.59%**  | 1.31  |
| 3      | 23.46%  | 7.14%   | 20.01%  | 10.95%  | 12.08%     | **9.47%**  | 1.33  |
| 3      | 19.45%  | 5.24%   | 23.24%  | 8.80%   | 9.98%      | **7.01%**  | 1.34  |
| 4      | 27.22%  | 5.61%   | 39.16%  | 11.24%  | 18.47%     | **9.12%**  | 1.63  |
| 5      | 13.87%  | 7.53%   | 64.62%  | 13.32%  | 14.62%     | **10.81%** | 1.43  |
| 5      | 17.54%  | 8.86%   | 59.21%  | 11.83%  | 29.39%     | **10.74%** | 1.21  |
| 5      | 48.57%  | 9.50%   | 41.93%  | 22.14%  | 28.16%     | **18.22%** | 1.92  |
| 5      | 54.92%  | 8.88%   | 36.77%  | 24.23%  | 24.88%     | **21.25%** | 2.39  |
| 6      | 50.47%  | 10.10%  | 48.96%  | 25.51%  | 33.19%     | **20.58%** | 2.04  |
| 8      | 42.67%  | 7.89%   | 69.92%  | 30.60%  | 47.78%     | **20.45%** | 2.59  |
| 9      | 39.56%  | 7.91%   | 99.27%  | 16.90%  | 17.99%     | **14.09%** | 1.78  |

Table 4: Bounding the DCP for multiclass classifiers with known confusion matrices; Natality data set with a $k$-Nearest-Neighbor classifier. Table columns as in Table 1.

| Test        | Lower  | Upper Bounds | | | | Best  |
|-------------|--------|---------|---------|------------|-----------|-------|
|             | Bound  | Average | Greedy  | Average+LM | Greedy+LM | Ratio |
| Attendant   | 1.80%  | 28.53%  | **1.82%** | **1.82%**  | **1.82%**  | 1.01  |
| Father Race | 1.17%  | 46.54%  | 1.20%   | **1.18%**  | **1.18%**  | 1.01  |
| Mother Race | 0.61%  | 21.47%  | 0.69%   | **0.62%**  | **0.62%**  | 1.02  |
| Payer       | 1.73%  | 54.61%  | 1.79%   | 1.76%      | **1.75%**  | 1.01  |

Table 5: Bounding the DCP for multiclass classifiers with known confusion matrices; US Census data set with the neural network-based classifiers. Table columns as in Table 1.

| # Labels | Error | Lower Bound | Upper Bounds | | | | Best Ratio |
|---|---|---|---|---|---|---|---|
| | | | Average | Greedy | Average+LM | Greedy+LM | |
| 3 | 12.76% | 2.36% | 20.19% | 3.46% | 3.04% | **3.03%** | 1.28 |
| 3 | 12.12% | 1.43% | 23.16% | 2.06% | 2.75% | **1.84%** | 1.28 |
| 3 | 8.95% | 1.27% | 18.89% | 2.05% | 1.75% | **1.75%** | 1.38 |
| 3 | 8.96% | 1.52% | 16.63% | 1.94% | 1.83% | **1.77%** | 1.16 |
| 3 | 6.04% | 2.42% | 22.88% | 3.52% | **2.96%** | **2.96%** | 1.22 |
| 3 | 8.26% | 0.86% | 34.80% | 3.53% | **2.24%** | **2.24%** | 2.60 |
| 3 | 13.31% | 0.01% | 68.05% | 27.12% | **0.01%** | 0.02% | 2.45 |
| 3 | 8.07% | 0.63% | 24.58% | 1.41% | 0.79% | **0.78%** | 1.23 |
| 3 | 10.39% | 1.53% | 40.93% | 1.56% | 2.06% | **1.56%** | 1.02 |
| 4 | 4.08% | 0.93% | 41.29% | 4.07% | 2.51% | **2.17%** | 2.33 |
| 5 | 4.09% | 0.78% | 69.65% | 12.97% | 1.89% | **0.99%** | 1.28 |
| 5 | 5.40% | 2.44% | 66.73% | 41.57% | 3.18% | **2.53%** | 1.04 |
| 5 | 17.48% | 3.55% | 46.89% | 4.35% | 4.24% | **4.05%** | 1.14 |
| 5 | 21.35% | 6.02% | 64.82% | 8.01% | 8.86% | **7.67%** | 1.27 |
| 6 | 17.76% | 2.49% | 66.61% | 6.01% | 4.30% | **3.67%** | 1.47 |
| 8 | 26.69% | 2.64% | 68.10% | 14.22% | 5.66% | **4.28%** | 1.62 |
| 9 | 18.19% | 2.95% | 80.76% | 3.13% | 4.15% | **3.12%** | 1.06 |

Table 6: Bounding the DCP for multiclass classifiers with known confusion matrices; Natality data set with the neural network classifier. Table columns as in Table 1.

| Protected Attribute | Lower Bound | Upper Bounds | | | | Best Ratio |
|---|---|---|---|---|---|---|
| | | Average | Greedy | Average+LM | Greedy+LM | |
| Attendant | 1.03% | 26.25% | 1.05% | **1.04%** | **1.04%** | 1.01 |
| Father Race | 0.67% | 50.53% | 0.73% | **0.67%** | **0.67%** | 1.01 |
| Mother Race | 0.58% | 13.25% | 0.58% | **0.58%** | **0.58%** | 1.01 |
| Payer | 1.21% | 55.86% | 1.21% | **1.21%** | **1.21%** | 1.00 |

Table 7: Comparing the output of the minDCP local optimizer (LO) to the DCP range calculated for US Census classifiers for the neural network classifiers. The ranges are derived from Table 5.

| US Census | Neural Network | |
|:---:|:---:|:---:|
| # Labels | minDCP LO | true DCP |
| 3 | 0.97% | $2.36\% - 3.03\%$ |
| 3 | 0.63% | $1.43\% - 1.84\%$ |
| 3 | 0.68% | $1.27\% - 1.75\%$ |
| 3 | 0.53% | $1.52\% - 1.77\%$ |
| 3 | 1.23% | $2.42\% - 2.96\%$ |
| 3 | 0.56% | $0.86\% - 2.24\%$ |
| 3 | 0.01% | 0.01% |
| 3 | 0.26% | $0.63\% - 0.78\%$ |
| 3 | 0.90% | $1.53\% - 1.56\%$ |
| 4 | 0.47% | $0.93\% - 2.17\%$ |
| 5 | 0.70% | $0.78\% - 0.99\%$ |
| 5 | 1.36% | $2.44\% - 2.53\%$ |
| 5 | 1.09% | $3.55\% - 4.05\%$ |
| 5 | 3.16% | $6.02\% - 7.67\%$ |
| 6 | 0.80% | $2.49\% - 3.67\%$ |
| 8 | 1.77% | $2.64\% - 4.28\%$ |
| 9 | 1.63% | $2.95\% - 3.12\%$ |

Table 8: Comparing the output of the minDCP local optimizer (LO) to the DCP range calculated for the Natality data set for the neural network classifier. The ranges are derived from Table 6.

| Natality | Neural Network | |
|:---:|:---:|:---:|
| Protected Attribute | minDCP LO | true DCP |
| Attendant | 0.15% | $1.03\% - 1.04\%$ |
| Father Race | 0.24% | 0.67% |
| Mother Race | 0.10% | 0.58% |
| Payer | 0.13% | 1.21% |

Table 9: Calculated `minDCP` local optimizers for the UK elections data. In each line, the election data from the listed baseline year was used to predict the vote in the listed prediction year.

| Election years | | # protected attribute values | `minDCP` LO |
|---|---|---|---|
| Baseline | Prediction | | |
| 1918 | 1922 | 12 | 6.36% |
| 1922 | 1923 | 13 | 8.57% |
| 1923 | 1924 | 13 | 3.26% |
| 1924 | 1929 | 13 | 4.92% |
| 1929 | 1931 | 13 | 5.14% |
| 1931 | 1935 | 13 | 3.11% |
| 1935 | 1945 | 13 | 5.69% |
| 1945 | 1950 | 12 | 4.75% |
| 1950 | 1951 | 12 | 3.79% |
| 1951 | 1955 | 12 | 5.54% |
| 1955 | 1959 | 12 | 2.40% |
| 1959 | 1964 | 12 | 2.16% |
| 1964 | 1966 | 12 | 1.37% |
| 1966 | 1970 | 12 | 1.32% |
| 1970 | 1974 (Feb) | 12 | 2.69% |
| 1974 (Feb) | 1974 (Oct) | 12 | 6.28% |
| 1974 (Oct) | 1979 | 12 | 3.67% |
| 1979 | 1983 | 11 | 3.23% |
| 1983 | 1987 | 12 | 1.43% |
| 1987 | 1992 | 12 | 4.21% |
| 1992 | 1997 | 11 | 5.25% |
| 1997 | 2001 | 12 | 5.25% |
| 2001 | 2005 | 12 | 4.83% |
| 2005 | 2010 | 12 | 5.26% |
| 2010 | 2015 | 12 | 3.96% |
| 2015 | 2017 | 12 | 5.54% |
| 2017 | 2019 | 11 | 5.24% |

## B. An Additional Experiment

In this experiment, we studied a data set on US education (USDA Economic Research Service, 2021), which provides the percentage of various levels of education attainment (e.g., high school, college) in each US state in each decade. Here too, we calculated `minDCP` for a hypothesized classifier that predicts the education level to be distributed the same in each state in each decade. The protected attribute as set to be the state. Table 10 provides our results. Here, we found no significant differences in the DCP of change patterns in different decades, indicating a fairly constant behavior of this measure of divergence between states. This type of analysis can be used for exploratory research on social questions.

Table 10: Calculated `minDCP` upper bounds for the US education data set.

| Year | | |
| --- | --- | --- |
| Baseline | Predicted | `minDCP` upper bound |
| 1970 | 1980 | 2.38% |
| 1980 | 1990 | 2.94% |
| 1990 | 2000 | 2.22% |
| 2000 | 2015-2019 | 2.32% |

## C. The Greedy Initialization Procedure

We provide here the full details of the greedy initialization procedure presented in Section 5.

Let $\mathbf{f} := \{f_y\}_{y \in \mathcal{Y}}$, where $f_y : \mathcal{Y} \to \mathcal{Y}$, be label mappings conditioned on the true label $y$, which can map some predicted labels to the same transformed label. For a given classifier $\mathcal{C}$ with distribution $\mathcal{D}$, let $\mathcal{C}[\mathbf{f}]$ be a hypothetical classifier that predicts $f_y(\hat{y})$ whenever the true label is $y$ and $\mathcal{C}$ would predict $\hat{y}$. For a given distribution $\mathcal{P}$ over $(Y, \hat{Y}, A)$, let $\mathcal{P}[\mathbf{f}]$ be the distribution of $(Y, f_Y(\hat{Y}), A)$. Then, $\mathcal{D}[\mathbf{f}]$ is the distribution determined by $\mathcal{C}[\mathbf{f}]$. It is easy to see that $\mathrm{DCP}(\mathcal{C}[\mathbf{f}]) \leq \mathrm{DCP}(\mathcal{C})$. This is because the equality $\mathcal{D}_a^y = (1 - \eta_a^y)\mathcal{D}_\mathsf{b}^y + \eta_a^y \mathcal{N}_a^y$ implies that also $\mathcal{D}_a^y[\mathbf{f}] = (1 - \eta_a^y)\mathcal{D}_\mathsf{b}^y[\mathbf{f}] + \eta_a^y \mathcal{N}_a^y[\mathbf{f}]$. Thus, minimizing over $\eta_a^y$ for $\mathcal{C}[\mathbf{f}]$ can never result in a solution of a higher value than minimizing for $\mathcal{C}$. We use this observation to devise an iterative greedy optimization procedure.

For $i \in [k-1]$, let $\mathbf{f}_i := (f_{i,y})_{y \in \mathcal{Y}}$ be an indexed set of label mappings, $f_{i,y} : \mathcal{Y} \to \mathcal{Y}$, defined as follows. Let $y_i$ be the $i$'th label in $\mathcal{Y}$ that is different from $y$. Denote $\mathcal{Y}_i = \{y, y_1, \ldots, y_i\}$. Note that $\mathcal{Y}_{k-1} = \mathcal{Y}$. For $i \in [k-2]$, define $f_{i,y}(j) = j \cdot \mathbb{I}[j \in \mathcal{Y}_i] + y_i \cdot \mathbb{I}[j \notin \mathcal{Y}_i]$. Note that $f_{i,y}$ can be calculated from the image of $f_{i-1,y}$. Hence, $\mathcal{C}[\mathbf{f}_i]$ is a refinement of $\mathcal{C}[\mathbf{f}_{i-1}]$. In addition, $f_{k-1,y}$ is the identity. Thus, the following monotonicity property holds:

$$\mathrm{DCP}(\mathcal{C}[\mathbf{f}_1]) \leq \mathrm{DCP}(\mathcal{C}[\mathbf{f}_2]) \leq \ldots \leq \mathrm{DCP}(\mathcal{C}[\mathbf{f}_{k-1}]) = \mathrm{DCP}(\mathcal{C}).$$

Moreover, $\mathrm{DCP}(\mathbf{f}_1)$ can be calculated exactly as in case of binary classification, since the range of $\mathbf{f}_1$ includes only $y$ and $y_1$. Based on these observations, we define a greedy procedure for calculating an assignment for $\boldsymbol{\alpha}_\mathsf{b}^y$ to initialize the minimization in Eq. (5).

Let $\boldsymbol{\alpha}_\mathcal{A}^y[i]$ be row $y$ of the confusion matrices of $\mathcal{D}[\mathbf{f}_i]$. Then coordinates $j \in \mathcal{Y}_{i-1}$ of $\boldsymbol{\alpha}_\mathcal{A}^y[i]$ are the same as those of $\boldsymbol{\alpha}_\mathcal{A}^y$, and coordinate $y_i$ has the value $\widetilde{\alpha}_a^{yy_i} := \sum_{j=i}^{k-1} \alpha_a^{yy_j}$. We have

$$\mathrm{DCP}_y(\boldsymbol{\alpha}_\mathcal{A}^y[1]) \leq \mathrm{DCP}_y(\boldsymbol{\alpha}_\mathcal{A}^y[2]) \leq \ldots \leq \mathrm{DCP}_y(\boldsymbol{\alpha}_\mathcal{A}^y[k-1]) = \mathrm{DCP}_y(\boldsymbol{\alpha}_\mathcal{A}^y).$$

The greedy procedure first calculates an assignment for $\boldsymbol{\alpha}_\mathsf{b}^y[1]$ that obtains the value of $\mathrm{DCP}_y(\boldsymbol{\alpha}_\mathcal{A}^y[1])$. This is a binary problem, which can be solved exactly following Sabato & Yom-Tov (2020). Then, at each iteration $i+1$ for $i \in [k-2]$, a local minimum $\boldsymbol{\alpha}_a^y[i+1]$ for $\mathrm{DCP}_y(\boldsymbol{\alpha}_\mathcal{A}^y[i+1])$ is calculated by constraining $\boldsymbol{\alpha}_a^y[i+1]$ to have the same coordinates as $\boldsymbol{\alpha}_\mathsf{b}^y[i]$ on $\mathcal{Y}_{i-1}$, and minimizing over $\alpha_\mathsf{b}^{yy_i}, \alpha_\mathsf{b}^{yy_{i+1}}$ such that their sum is equal to coordinate $y_i$ in $\boldsymbol{\alpha}_\mathsf{b}^y[i]$. This minimization can be solved exactly, as follows.

Denote the value of coordinate $y_i$ in $\boldsymbol{\alpha}_\mathsf{b}^y[i]$ by $\gamma = 1 - \sum_{j \in \mathcal{Y}_{i-1}} \alpha_\mathsf{b}^{yj}$. Minimizing the objective of $\mathrm{DCP}_y(\boldsymbol{\alpha}_\mathcal{A}^y[i+1])$ subject

to the constraints resulting from $\boldsymbol{\alpha}_{\mathsf{b}}^{y}[i]$ is equivalent solving the following problem:

$$\underset{\alpha_{\mathsf{b}}^{yy_i}, \alpha_{\mathsf{b}}^{yy_{i+1}}}{\text{Minimize}} \sum_{a \in \mathcal{A}} w_a \pi_a^y \max\{\max_{\hat{y} \in \mathcal{Y}_i} \eta(\alpha_{\mathsf{b}}^{y\hat{y}}, \alpha_a^{y\hat{y}}), \eta(\alpha_{\mathsf{b}}^{yy_{i+1}}, \widetilde{\alpha}_a^{yy_{i+1}})\}$$

$$\text{s.t.} \qquad \alpha_{\mathsf{b}}^{yy_i}, \alpha_{\mathsf{b}}^{yy_{i+1}} \geq 0 \text{ and } \alpha_{\mathsf{b}}^{yy_i} + \alpha_{\mathsf{b}}^{yy_{i+1}} = \gamma.$$

Letting $v_a := \max_{\hat{y} \in \mathcal{Y}_{i-1}} \eta(\alpha_{\mathsf{b}}^{y\hat{y}}, \alpha_a^{y\hat{y}})$, this is equivalent to

$$\underset{\alpha_{\mathsf{b}}^{yy_i} \in [0, \gamma]}{\text{Minimize}} \sum_{a \in \mathcal{A}} w_a \pi_a^y \max\{v_a, \eta(\alpha_{\mathsf{b}}^{yy_i}, \alpha_a^{yy_i}), \eta(\gamma - \alpha_{\mathsf{b}}^{yy_i}, \widetilde{\alpha}_a^{yy_{i+1}})\}.$$

Similarly to the case of binary classification, this objective is one-dimensional, and concave in each of the intervals defined by the inflection points of the $\eta$ instances and the values for which any two of the expressions in the maximum are equal. Denote this set of points by $M_i^y$. Then the objective above is minimized by one of the values in the following set: $M_i^y \cup \{\alpha_a^{yy_i}\}_{a \in \mathcal{A}} \cup \{\widetilde{\alpha}_a^{yy_{i+1}}\}_{a \in \mathcal{A}} \cup \{0, \gamma\}$. Repeating this procedure until iteration $i = k - 1$, we obtain an assignment for $\boldsymbol{\alpha}_{\mathsf{b}}^{y}$ which can be used to calculate an upper bound for $\text{DCP}_y(\boldsymbol{\alpha}_{\mathcal{A}}^{y})$.

Since the ordering of the labels in the greedy procedure is arbitrary, it is possible to attempt several different orderings and select the one that obtains the smallest DCP value. In our experiments, we tried 10 random orderings in each upper bound calculation.

## D. More Details on the Local Optimization Procedure

Here we provide more details on the minimization of Eq. (15). Given the split in Eq. (16), the last constraint in Eq. (15) becomes

$$\eta(\alpha_{\hat{y}}^{y}, h_{a,\hat{y}}^{y}) \leq c_a^y \iff \left\{ \begin{array}{l} \eta_1(\alpha_{\hat{y}}^{y}, h_{a,\hat{y}}^{y}) \leq c_a^y \\ \eta_2(\alpha_{\hat{y}}^{y}, h_{a,\hat{y}}^{y}) \leq c_a^y \end{array} \right.,$$

and now the constraints does not include singularity points, and can be locally approximated by two linear functions, one for $\eta_1$ and one for $\eta_2$ (see Figure 2). Explicitly, the Taylor approximation of $\eta_i$ for $i = 1, 2$ are given by:

$$\eta_i(\alpha + \epsilon_a, b + \epsilon_b) \approx \eta_i(\alpha, b) + \frac{\partial \eta_i}{\partial \alpha} \epsilon_\alpha + \frac{\partial \eta_i}{\partial b} \epsilon_b,$$

where $\frac{\partial \eta_1}{\partial \alpha} = \frac{b}{\alpha^2}$, $\frac{\partial \eta_1}{\partial b} = -\frac{1}{\alpha}$, $\frac{\partial \eta_2}{\partial \alpha} = \frac{-(1-b)}{(1-\alpha)^2}$, and $\frac{\partial \eta_2}{\partial b} = \frac{1}{1-a}$.

Given the first order Taylor approximations above, we form the LP approximation of Eq. (15) around an iterate $\mathbf{x}^{(t)} = \{\widetilde{\boldsymbol{\alpha}}^{y}, \widetilde{\mathbf{H}}^{y}, \widetilde{\mathbf{c}}^{y}\}_{y=1}^{k}$ as follows:

$$\underset{\{\boldsymbol{\alpha}^y\}, \{\mathbf{H}^y\}, \{\mathbf{c}^y\}}{\text{Minimize}} \sum_{y} \langle \widetilde{\mathbf{w}}^y, \mathbf{c}^y \rangle \tag{17}$$

$$\begin{aligned}
\text{s.t.} \quad & 0 \leq \boldsymbol{\alpha}^y \leq 1 \quad \forall y \in [k], \\
& 0 \leq \mathbf{H}^y \leq 1 \quad \forall y \in [k], \\
& 0 \leq \mathbf{c}^y \leq 1 \quad \forall y \in [k], \\
& \langle \boldsymbol{\alpha}^y, \mathbf{1}_k \rangle = 1 \quad \forall y \in [k], \\
& \mathbf{H}^y \mathbf{1}_k = \mathbf{1}_{|\mathcal{A}|} \quad \forall y \in [k], \\
& \sum_{y=1}^{k} \pi_a^y h_{a,\hat{y}}^y = p_a^{\hat{y}} \quad \forall \hat{y} \in [k], a \in [|\mathcal{A}|]. \\
& \eta_i(\widetilde{\alpha}_{\hat{y}}^{y}, \widetilde{h}_{a,\hat{y}}^{y}) + \frac{\partial \eta_i}{\partial \alpha}(\alpha_{\hat{y}}^{y} - \widetilde{\alpha}_{\hat{y}}^{y}) + \frac{\partial \eta_i}{\partial b}(h_{a,\hat{y}}^{y} - \widetilde{h}_{a,\hat{y}}^{y}) \leq c_a^y, \quad \forall y, \hat{y} \in [k], \ a \in [|\mathcal{A}|], i = 1, 2,
\end{aligned}$$

Eq. (17) is an LP problem which is solved at each iteration by an LP solver, for which we use the `scipy.optimize` library. In addition to the box constraints of $[0, 1]$ for all variables, it includes $2 \cdot k \cdot |\mathcal{A}| + k$ equality constraints, and $2 \cdot k^2 \cdot |\mathcal{A}|$ inequality constraints. It can be seen that the number of inequality constraints is rather high, and as a result, so is the computational complexity of the algorithm if run as is. However, many of these constraints are not active in the solution, and in any case we limit the step size of our algorithm. Hence, we can ease the difficulty of the LP problem by

both limiting the search space of the LP solver, and removing the inequality constraints that seem to be inactive in the solution. To this end, we remove the inequality constraints where $\eta_i(\tilde{\alpha}_{\hat{y}}^y, \tilde{h}_{a,\hat{y}}^y) < -1$, as we expect these not to be active at the solution. Furthermore, we use a maximum step size $\tau$ so that $\|\boldsymbol{\alpha}^y - \tilde{\boldsymbol{\alpha}}^y\|_\infty < \tau$, and $\|\mathbf{H}^y - \tilde{\mathbf{H}}^y\|_\infty < \tau$, where $\|\cdot\|_\infty$ is the maximum norm. This condition is trivially incorporated in the box constraints of Eq. (17). Finally, given the LP approximation above, the solver of Eq. (15) follows the same lines as Alg. 1.

As in Section 6, numerical instabilities arise when $\alpha_a^{y\hat{y}}$ is too close to 0 or 1. Thus, here as well we restrict these values to be in the segment $[\varepsilon, 1 - \varepsilon]$, this time as optimization variables through the box constraints. Also, we update the values $p_a^{\hat{y}}$ to be $p_a^{\hat{y}} = (1 - k\epsilon)p_a^{\hat{y}} + \epsilon$ to guarantee a solution for Eq. (15) after updating the box constraints for $\alpha_a^{y\hat{y}}$.

**Convergence of Algorithm 1**    Our problem contains the non-linear functions $\eta$, which are approximated by linear functions. For a general non-linear programming problem, sequential linear programming (SLP) methods converge linearly if all functions are smooth. That is because they do not use Hessian information, and their whole purpose is to handle the constraints efficiently (determine who is active and who is not). If one uses Hessian information, we get sequential quadratic programming (SQP) and the convergence is asymptotically quadratic once the active set of constraints is identified, similar to Newton's method (Nocedal & Wright, 2006). For an SLP method like in Algorithm 1 to converge linearly, we first need to have smooth constraints, and for that we use the split of presented in Eq. (16) and detailed above. Furthermore, we also limit the size of the steps to be of size at most $\tau$. This approach is called a Trust Region method, and is commonly used with sequential quadratic or linear programming methods. The method was analyzed in (Kiessling et al., 2022) for a case that is similar to ours, where the objective of the constrained problem is linear, like in Eq. (9) and Eq. (15). It is shown that the convergence of the SLP method is linear, and the rate depends on the radius of the trust region method ($\tau$, in our case). In Figure 4 we show the convergence of our method for solving Eq. (15) for a convex $\eta$, so that the problem is convex and the solutions that are obtained by the method using all trust regions are equivalent. Specifically, to demonstrate the convergence we used

$$\eta_{convex}(a, b) = \begin{cases} a^2 - ab & b < a, \\ (1 - a)^2 - (1 - b)(1 - a) & b > a, \\ 0 & b = a, \end{cases} \tag{18}$$

instead of Eq. (4), and got the plots in Figure 4 for various maximal step sizes $\tau$, for solving Eq. (15) for the labor dataset. Here, the plots show that the method is slower as $\tau$ is smaller, since the algorithm is constrained to be slower. However, in some cases, the inner LP solution can result in a stagnate direction, and then decreasing $\tau$ as shown in (Kiessling et al., 2022) can mitigate this. In our original problem, $\eta$ and the whole problem are non-convex, hence the algorithm can converge to a different local minimum for each value of $\tau$. For the problems reported in this paper, we used $\tau = 0.2$ which seemed to work best, and decreased $\tau$ when the LP solver failed to converge.

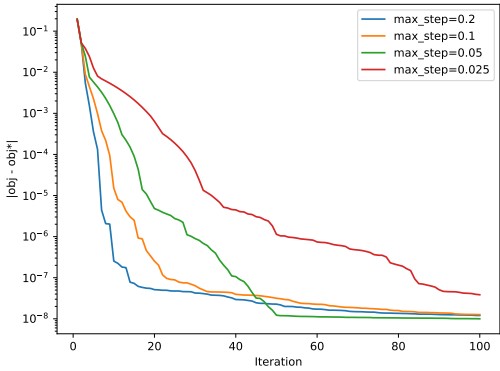

Figure 4: The convergence of Algorithm 1 for different trust region (maximal step size) parameters $\tau$.

