# OpenReview forum: "Disparate Conditional Prediction in Multiclass Classifiers"
_ICML.cc/2025/Conference — ICML 2025 poster_

### Official Review · Reviewer_PQXZ · 2025-03-09

**Overall Recommendation:** 3

**Summary:**

This paper introduced the measure of Disparate Conditional Prediction (DCP) to multi-class classifiers based on equalized odds. They provided optimization methods for scenarios with and without the access to confusion matrices and justified the algorithm efficacy by using decision trees and NN models on three datasets.

**Claims And Evidence:**

DCP is extended to multi-class scenarios, and I think the claims are well evidenced in the paper.

**Essential References Not Discussed:**

This work is based on Sabato&Yom-Tov (2020) and I think the most related works are included in the paper.

**Experimental Designs Or Analyses:**

The results comparisons are among different strategies or with the true DCP. It would better to simply declare this is the best choice if there is not any feasible baselines.

**Methods And Evaluation Criteria:**

The proposed DCP measure looks good, and some standard datasets are applied.

**Other Comments Or Suggestions:**

1. In Eq. 4, both w and \pi are not dependent on the classifier according to their definitions in section 3, but they are said determined by C below Eq. 3.
2. Eq. 2 executes addition group by group, then will it lead to a trivial solution?

**Other Strengths And Weaknesses:**

Strengths: The proposed DCP for multi-class scenarios is well presented, and I believe some readers in the fairness community will find this work interesting. Although I may not fully grasp every detail, I find the paper to be of good quality.

Weaknesses:
1. Some figures are presented without further explanation. For example, what we are expected to see or understand through Figure 1. The authors should not just simply refer to it (line 146).
2. Some notations are not easy to understand. For example, above Eq. 2 we see \eta’s definition, while it takes two input variables in Eq. 4I found this part confusing when reading here.

**Questions For Authors:**

1. Equalized odds emphasizes true and false positive rates, which are well-interpreted in binary classification. It would be helpful if the authors could provide some examples illustrating the significance of equalized odds in multi-class scenarios.
2. I understand this work currently focuses on auditing a trained classifier. But I am curious have you considered integrating the proposed DCP into a in-processing fairness method? What is the challenge of doing so?

**Relation To Broader Scientific Literature:**

I am unsure whether this topic holds significant importance in fairness research today. Maybe more fairness audit research should be discussed to better position this work.

**Theoretical Claims:**

I read through the theoretical analyses but was unable to verify their correctness.

---

> ### Author Rebuttal · Authors · 2025-03-31
>
> Thank you for your helpful review and comments. Please find below our response to your comments and questions.
>
> **Comment**: The results comparisons are among different strategies or with the true DCP. It would be better to simply declare this is the best choice if there are not any feasible baselines.
>
> **Reply**: We have included different strategies for ablation purposes, as we believe it strengthens the justification of our use of the chosen strategies.
>
> **Comment**: Maybe more fairness audit research should be discussed to better position this work.
>
> **Reply**: Our work falls within the realm of fairness auditing using only aggregate statistics, without assuming access to the classifier. This is in contrast with many other works, which require access to the classifier and the ability to query it. The challenge of auditing fairness using limited information has received significant attention in recent years, as evident, for example, in [1] & [2]. Our work is unique, in that it is the first, to our knowledge, to address the multiclass setting.
>
> The state of fairness audit research is discussed in the first three paragraphs of the Introduction, and in the second paragraph of the Related Work section. We will expand the discussion on recent works on fairness auditing with limited information in the final version of the paper.
>
> [1] Pinzón, Carlos, Catuscia Palamidessi, Pablo Piantanida, and Frank Valencia. "On the incompatibility of accuracy and equal opportunity." Machine Learning 113, no. 5 (2024): 2405-2434.
>
> [2] Jialu Wang, Yang Liu, and Caleb Levy. Fair classification with group-dependent label noise. In Proc. of the 2021 ACM Conference on Fairness, Accountability, and Transparency, FAccT ’21, page 526–536, New York, NY, USA, 2021b. Association for Computing Machinery.
>
>
> **Comment**: What we are expected to see or understand through Figure 1.
>
> **Reply**: Figure 1 provides a visualization of the function $\eta$, demonstrating that it is piecewise concave. We will refer to the figure when discussing the function's properties and explain the visualization.
>
> **Comment**: Above Eq. 2 we see $\eta$’s definition, while it takes two input variables in Eq. 4.
>
> **Reply**: Above and in Eq. 2, we use $\eta_a^y$, which is a scalar, and $\eta_a$, which is a vector of scalars. $\eta$ without superscripts or subscripts is a function that takes two arguments. We will clarify this in the final version.
>
> **Numbered Comment 1**: In Eq. 4, both w and $\pi$ are not dependent on the classifier according to their definitions in section 3, but they are said determined by C below Eq. 3.
>
> **Reply**: Thank you for the comment. Indeed this is a typo, since only $M_A$ is determined by C. We will fix this in the final paper.
>
> **Numbered Comment 2**: Eq. 2 executes addition group by group, then will it lead to a trivial solution?
>
> **Reply**: As our analysis below Eq (2) shows, the solution is not trivial, because of the constraint that the values of $\eta_a^y$ must be consistent with the distribution D.
>
> **Question 1**: Equalized odds emphasizes true and false positive rates, which are well-interpreted in binary classification. It would be helpful if the authors could provide some examples illustrating the significance of equalized odds in multi-class scenarios.
>
> **Reply**: In multiclass scenarios, the multiclass equalized odds criterion measures any differences in conditional prediction probabilities between sub-populations. This includes not only the difference in the rate of correct predictions as in the binary case, but also the types of prediction mistakes. For instance, if a patient's heart attack is misdiagnosed as an anxiety attack (which may mean the patient is denied care), this is significantly different than being misdiagnosed as a stroke (which may lead to delayed care). If some sub-populations incur more of a certain type of misdiagnosis error, this could indicate bias in diagnosis, as well as lead to undesired differences in treatment.  This is one example of the importance of the multiclass equalized odds criterion. We will add a discussion with additional examples to the final version of the paper.
>
> **Question 2**: I understand this work currently focuses on auditing a trained classifier. But I am curious have you considered integrating the proposed DCP into a in-processing fairness method?
>
> **Reply**: We agree that this is an important question, and we intend to study it in future work.

---

### Official Review · Reviewer_1f59 · 2025-03-11

**Overall Recommendation:** 3

**Summary:**

The authors proposed a multi label extension of the framework developed by Sabato & Yom-Tov (thats only binary) that allows to compute bounds on fairness metrics only using population level quantities.

Dcp is determined by the matrix, and the proportions and a function of conditional prediction rates and such that there is an efficient way on computing it, the authors then derive a multiclass version of DCP. Then the authors compute an analytical lower bound on the DCP and then a procedure to upper bound the multiclass version of the DCP. This procedure is a loop of LPs which can be solver efficiently. Finally the authors present a way to compute a bound on the DCP with out the need of having all the conditional proportions (the per class matrix), in particular there is no access to the classifier, just to its population level frequencies.

Weakness
- The writing of the paper is confusing. The derivation of the multiclass DCP should be a theroem. Also the metric DCP lacks motivation.

Strengths
-  The authors propose an efficient algorithm to compute bounds on DCP.
- The algorithms can be computed efficiently

**Claims And Evidence:**

Yes, the experiments and proved are correct and relevant.

**Essential References Not Discussed:**

Refer to summary

**Experimental Designs Or Analyses:**

Refer to summary

**Methods And Evaluation Criteria:**

They do, but the DCP is not propperly motivated

**Other Comments Or Suggestions:**

...

**Other Strengths And Weaknesses:**

Refer to summary

**Questions For Authors:**

No questions for the authors.

**Relation To Broader Scientific Literature:**

I am somehow familiar with the fairness literature but not with the main work the authors cite (Sabato and Yom-tov)

**Theoretical Claims:**

Refer to summary

---

> ### Author Rebuttal · Authors · 2025-03-31
>
> Thank you for your helpful review and comments. Please find below our response to your comments and questions.
>
> **Comment**: The derivation of the multiclass DCP should be a theorem.
>
> **Reply**: Thank you for the suggestion. We will add a formal theorem statement for the derivation of DCP.
>
> **Comment**: the metric DCP lacks motivation.
>
> **Reply**: The main motivation of the DCP, as discussed in the Introduction, is that it has a consistent interpretable meaning as a fraction of the population, regardless of the number of protected attribute values, the number of classes, or the degree of class imbalance. Thus, DCP is useful for interpretably auditing and comparing classifiers. We will add a discussion of the disadvantages of other previously suggested measures, to highlight the issues that are overcome by the use of DCP. These include the following:
>
> - Measures based on differences discount differences that are small in value, although they could be meaningful in terms of fairness. This is especially problematic when there is class imbalance. For instance, if the probability of predicting a certain label is 0.1 in one sub-population and 0.2 in another, this would be identified as significantly more unfair than if the probabilities are 0.001 and 0.002, respectively. However, when this label is very rare to begin with (such as in cancer diagnosis), these differences could indicate meaningful unfairness. Previous measures that try to correct this, such as ratio-based measures, suffer from other issues, such as lack of boundedness and difficulty achieving normalization that retains the meaning of the measure. The DCP measure does not have these issues, since it always identifies the fraction of the population that is affected by the disparate classification probabilities, which is inherently normalized and meaningful, regardless of class imbalance.
>
> - Most previous works only consider cases with two sub-populations. Standard extensions to multiple sub-populations maximize over pairwise comparisons between sub-populations. However, this disproportionally penalizes classifiers in which a single sub-population is treated unfairly, in comparison to classifiers in which several sub-populations are treated unfairly, since both types of classifiers would be deemed to have the same amount of unfairness. The DCP measure uses an optimal common baseline and sums over all sub-population differences from the baseline, thus properly differentiating between classifiers of significantly different fairness levels.
>
> - The lack of quantifiable interpretability of the measures proposed in the literature implies that they are not guaranteed to have a consistent interpretation of unfairness across all possible classifiers and confusion matrices. While we have listed above several specific issues, trying to solve these issues by fixing the measure in an ad-hoc manner leads to other issues, and so forth. The quantifiable interpretability of DCP ensures that the value of the measure is meaningful in all possible scenarios, and that this meaning is consistent, so that one can also compare the value of the measure for different classifiers and obtain meaningful conclusions.

---

### Official Review · Reviewer_PiSi · 2025-03-12

**Overall Recommendation:** 3

**Summary:**

This paper provides methods to bound the unfairness of multiclass classifiers. In particular, they extend the Disparate Conditional Prediction (DCP) metric from prior work to the multiclass setting. The DCP measure for a classifier quantifies the fraction of the population for whom the classifier's prediction distribution (conditional on the group and true outcome) differs from that of an equalized odds fair classifier. The paper shows how this metric can be generalized to the multiclass setting and forward methods to bound the DCP of any classifier with and without access to the group-conditional confusion matrices. They demonstrate the efficacy of this approach over multiple datasets, with the proposed evaluation approach achieving the sharpest bounds on the DCP of decision trees and kNNs trained on the datasets.

**Claims And Evidence:**

The main claim of the paper is that DCP is a useful metric for assessing the fairness of multiclass classifiers and can be evaluated effectively using the proposed methodology. Both claims seem well-justified.

For the former claim (on the usefulness of DCP), the introduction does a good job of motivating the necessity of a DCP-based method for multiclass classifier audits. However, there seems to be heavy reliance on prior works for this motivation (e.g.,  Sabato & Yom-Tov for DCP and  Wang et al. for limitations of other metrics in the multiclass setting). I would encourage the authors to expand the discussion/examples on the limitations of prior methods for multiclass classifier audit and the usefulness of DCP to make the paper more self-contained.

I am also curious if any of the limitations of the other metrics in multiclass setting can be demonstrated within the paper itself. For example, in the setting where one has access to the group confusion matrices, ${M_a}$, one can trivially construct quantify unfairness as $max_{a_1, a_2} || M_{a_1} - M_{a_2}||_F$, where $||\cdot||_F$ denotes the Frobenius norm. Its unclear whether the the main issue with this (and other prior ways of quantifying fairness for multiclass settings) is interpretabality of the metric or if they have fail to capture some crucial aspects of unfairness.

**Essential References Not Discussed:**

No

**Experimental Designs Or Analyses:**

The experimental design seems sound. However, Table 1, Table 2, and Figure 3 are quite difficult to read and interpret in their current form. I would encourage adding a more detailed caption and labels as well as increasing the font size of all tables and plots in the main body.

**Methods And Evaluation Criteria:**

The methods used to estimate DCP in the multiclass setting are well-described and look correct.

For the empirical analysis, most of the datasets used also make sense. However, I don't completely understand the purpose of the experiments on UK election patterns at the end of Section 7 and on US education data in the appendix. Specifically, it's unclear what the DCP criteria would imply in this case. The paper claims that a "high DCP value of such a classifier would indicate a possibly large difference between voting pattern changes across regions". However, it seems to me that any such inference can only be made when the classifier is fairly accurate and it's unclear if that's true in this case.

**Other Comments Or Suggestions:**

No other comments

**Other Strengths And Weaknesses:**

I found the Greedy initialization technique interesting. Table 1 shows that in some cases (e.g., when #labels=3) Greedy initialization alone leads to significant improvement in the upper bound, and running LM leads to a small improvement beyond that. In other cases (e.g., when #labels=8, 9), there is a larger gap between Greedy and Greedy+LM. I am curious if this is related to the number of labels or if its some artifact of how the greedy initialization is done.

**Questions For Authors:**

1. What are the limitations of the other metrics in multiclass setting and can these limitations be clearly demonstrated within the paper itself (e.g., using the datasets used for empirical analysis)?

2. What's the purpose of the experiments on UK election patterns and US education data and what does the DCP quantification mean in these settings?

3. Regarding the gap between the upper bounds from Greedy and Greedy+LM methods, what's the reason for the change in this gap across #labels? It seems this gap is smaller when #labels is small, but not sure if that trend will hold across datasets.

**Relation To Broader Scientific Literature:**

The paper adds to the literature on fairness audits methods. Specifically, it focuses on how to perform accurate fairness audits for multiclass classifiers. To do so, it employs the DCP measure from the prior work of Sabato & Yom-Tov (2020) and generalizes it to the multiclass setting.

To my knowledge, prior work on fairness hasn't focused much on interpretable extensions to multiclass settings. And the trivial extensions from binary to multiclass setting that first come to mind are of the type that I stated in the earlier section (using matrix norms), which may be easier to compute but not necessarily easy to interpret. In that sense, I like the formulation proposed in the paper and can it see being broadly relevant for audits of other unfair human/automated decision processes as well.

**Theoretical Claims:**

No theoretical claims.

---

> ### Author Rebuttal · Authors · 2025-03-31
>
> Thank you for your helpful review and comments. Please find below our response to your comments and questions.
>
> **Question 1**: What are the limitations of the other metrics in multiclass settings?
>
> **Reply**: Standard unfairness measures (see the Related Work section) have several disadvantages that are overcome by DCP. While these were discussed in the work that suggested binary DCP, we will add a relevant discussion to the paper, to highlight these issues and make the paper self-contained. The disadvantages include the following:
>
> - Measures based on differences discount differences that are small in value, although they could be meaningful in terms of fairness. This is especially problematic when there is class imbalance. For instance, if the probability of predicting a certain label is 0.1 in one sub-population and 0.2 in another, this would be identified as significantly more unfair than if the probabilities are 0.001 and 0.002, respectively. However, when this label is very rare to begin with (such as in cancer diagnosis), these differences could indicate meaningful unfairness. Previous measures that try to correct this, such as ratio-based measures, suffer from other issues, such as a lack of boundedness and difficulty achieving normalization that retains the meaning of the measure. The DCP measure does not have these issues, since it always identifies the fraction of the population that is affected by the disparate classification probabilities, which is inherently normalized and meaningful, regardless of class imbalance.
>
> - Most previous works only consider cases with two sub-populations. Standard extensions to multiple sub-populations maximize over pairwise comparisons between sub-populations. However, this disproportionally penalizes classifiers in which a single sub-population is treated unfairly, in comparison to classifiers in which several sub-populations are treated unfairly, since both types of classifiers would be deemed to have the same amount of unfairness. The DCP measure uses an optimal common baseline and sums over all sub-population differences from the baseline, thus properly differentiating between classifiers of significantly different fairness levels.
>
> - The lack of quantifiable interpretability of the measures proposed in the literature implies that they are not guaranteed to have a consistent interpretation of unfairness across all possible classifiers and confusion matrices. While we have listed above several specific issues, trying to solve these issues by fixing the measure in an ad-hoc manner leads to other issues, and so forth. The quantifiable interpretability of DCP ensures that the value of the measure is meaningful in all possible scenarios, and that this meaning is consistent, so that one can also compare the value of the measure for different classifiers and obtain meaningful conclusions.
>
> **Question 2**: What's the purpose of the experiments on UK election patterns and US education data and what does the DCP quantification mean in these settings?
>
> **Reply**:
> The experiments are intended to demonstrate how our approach can be used in diverse settings. Specifically, as discussed on page 8 of the paper, our analysis of the UK election dataset shows how political scientists could identify regional variability in voting patterns using the DCP measure. In the education dataset, our goal was to test whether educational attainment progressed more (or less) in some states compared to others. In this case, consider the scenario where authorities do not publish their achievements (or lack thereof) in disadvantaged areas, and only publish aggregate information. The DCP measure can help uncover such cases.
>
> **Question 3**: Regarding the gap between the upper bounds from Greedy and Greedy+LM methods, what's the reason for the change in this gap across #labels?
>
> **Reply**: This is indeed an interesting question. We conjecture that the greedy approach is less tight when there are more labels, because it relies on iterative solutions for a binary version of the problem. When the number of labels is larger, there are exponentially more possible combinations, and so we believe this leads to the greedy initialization being more likely to follow a sub-optimal iterative path.
>
> **Comment**: [On election patterns] it seems to me that any such inference can only be made when the classifier is fairly accurate.
>
> In the election experiment, the classifier predicts by assuming no changes in voting patterns between consecutive elections. Thus, we do not expect the classifier to always be accurate. On the contrary, its inaccuracy and unfairness are the objects of study, as they reveal changing voting patterns, and a variability of changes between regions, as measured by the DCP.
>
> **Comment**: Difficulty of interpreting Tables and a Figure.
>
> **Reply**: Thank you for the comment, we will improve the readability of these in the final version of the paper.

---

### Official Review · Reviewer_XH1z · 2025-03-15

**Overall Recommendation:** 4

**Summary:**

This work extends disparate conditional prediction, which measures the deviation from equalized odds, to multiclass classification problems. As the confusion matrices are not available, the authors derived the lower bound and the upper bound of the DCP of a multiclass classifier. The upper bounds are obtained using a local minimization procedure. To overcome the non-smoothness, the functions are split into locally linear parts, and the solutions are obtained by a combination of sequential solutions of standard linear programming problems.

**Claims And Evidence:**

The claims made in the submission are clearly supported by theoretical analysis on the lower bound and the upper bound.

**Essential References Not Discussed:**

No.

**Experimental Designs Or Analyses:**

In the experiments, only decision tree and nearest neighbor classifiers are examined. However, researchers would expect more complicated models, such as SVM and neural networks.

**Methods And Evaluation Criteria:**

The proposed methods look seamlessly reasonable.

**Other Comments Or Suggestions:**

The paper is clearly written. I did not find any typo.

**Other Strengths And Weaknesses:**

- The core contribution of this paper builds upon the criteria established by Sabato & Yom-Tov (2020), which raises questions about its originality and novel contributions.
- The approach to determining lower and upper bounds demonstrates sound optimization approach to addressing the problem.
- The experiments are only evaluated on simple tree-based and nearest neighbor models.

**Questions For Authors:**

- What does the nuisance distribution mean in Section 4? And what is its insight on real world data?
- What is the convergence rate of the proposed Algorithm 1?

**Relation To Broader Scientific Literature:**

The contribution of this paper is related to audit the fairness of multiclass classifiers.

**Theoretical Claims:**

There is no proof for the theoretical claims.

---

> ### Author Rebuttal · Authors · 2025-03-31
>
> Thank you for your helpful review and comments. Please find below our responses to your comments and questions.
>
> **Comment**: There is no proof for the theoretical claims.
>
> **Reply**: All theoretical claims are proved, either in the body of the paper or in the appendix. If there are any theoretical claims for which you believe proofs are missing, we would be grateful if you could point them out to us.
>
> **Comment**: In the experiments, only decision trees and nearest neighbor classifiers are examined. However, researchers would expect more complicated models, such as SVM and neural networks.
>
> **Reply**: The types of classifiers were chosen to be suitable for the low-dimensional categorical data-sets that we are reporting. Since our methods only use the prediction rates of each label in the output classifier, the complexity of the models does not have any effect on the usage of our methods. In addition, if a classifier is 100% accurate in its predictions, then the equalized odds criterion holds trivially, in which case the DCP would be zero. We appreciate the suggestion to extend our experiments - we will add experiments in which the classifiers are generated by SVMs and neural networks to the final version of the paper.
>
> **Comment**: originality and novel contributions
>
> **Reply**: While we build on a previous work, which defined binary DCP, the definitions and methods in that previous work are not applicable to multiclass classifiers, since they heavily rely on the binary nature of the problem. The calculation of the extension of DCP to the multiclass setting is challenging, and all of our methods for bounding the DCP are completely novel. As we discuss in the paper, the computational problems that arise in the multiclass case are not trivial and cannot be solved efficiently by out-of-the-box solvers. For instance, we use a combination of sequential linear programming with a function split to handle the non-smoothness. In addition, we handle instabilities encountered by the LP solvers when they deal with extremely high derivatives imposed by the $\eta$ function. Thus, our contributions are novel and original.
>
> **Question**: What does the nuisance distribution mean in Section 4? And what is its insight on real world data?
>
> **Reply**: The nuisance distribution represents the deviation of the classifier's conditional probabilities on each sub-population from the baseline conditional probabilities. This is a modeling construct that was used to derive the DCP measure in the previous work that we build upon. In this paper, we extend this to the multiclass case. In real-world data, a nuisance distribution with a higher mixture parameter ($\eta_a^y$) indicates that a larger part of the population is classified with conditional probabilities that deviate from the baseline conditional probabilities, thus indicating that the classifier is less fair.
>
> **Question**: What is the convergence rate of the proposed Algorithm 1?
>
> **Reply**: Our problem contains the non-linear $\eta$ functions, which are approximated by linear functions. For a general non-linear programming problem, sequential linear programming (SLP) methods converge linearly if all functions are smooth. That is because they do not use Hessian information, and their whole purpose is to handle the constraints efficiently (determine who is active and who is not). If one uses Hessian information, we get sequential quadratic programming (SQP) and the convergence is asymptotically quadratic once the active set of constraints is identified, similar to Newton’s method [1]. For Algorithm 1 to converge linearly, we first need to have smooth constraints, and for that we use the split of $\eta$ presented in Equation (16) and detailed in Appendix D. Furthermore, we also limit the size of the steps to be of size at most $\tau$, as detailed in Appendix D. This approach is called a Trust Region method, and is commonly used with sequential quadratic or linear programming methods. The method was analyzed in [2] for a case that is similar to ours, where the objective of the constrained problem is linear, like in Eq. (9) and Eq. (15). It is shown that the convergence of the SLP method is linear, and the rate depends on the radius of the trust region method ($\tau$, in our case). We will add this discussion to the final version of the paper, including plots demonstrating the convergence rate for different choices of $\tau$ in our problems.
>
> [1] Nocedal J, Wright SJ, editors. Numerical optimization. New York, NY: Springer New York; 1999.
>
> [2] Kiessling D, Zanelli A, Nurkanović A, Gillis J, Diehl M, Zeilinger M, Pipeleers G, Swevers J. A feasible sequential linear programming algorithm with application to time-optimal path planning problems. In 2022 IEEE 61st Conference on Decision and Control (CDC) 2022 Dec 6 (pp. 1196-1203). IEEE.

---

### Decision · Program_Chairs · 2025-05-01

**Decision:**

Accept (poster)

**Comment:**

The paper develops methods for auditing multiclass classifiers for equalized odds. A key contribution is generalizing Disparate Conditional Prediction (DCP) to multiclass prediction tasks. The authors develop methods for upper and lower bounding DCP.

The reviewers were unanimous in recommending acceptance. They found the paper well-written, well-motivated, and that the generalization of DCP to multiclass predictions is of interest to the community studying fairness in machine learning. I agree with this assessment, and recommend acceptance.